# In vitro CSC-derived cardiomyocytes exhibit the typical microRNA-mRNA blueprint of endogenous cardiomyocytes

Mariangela Scalise[1,8], Fabiola Marino[1,8], Luca Salerno[1], Teresa Mancuso[2], Donato Cappetta[3], Antonella Barone[1], Elvira Immacolata Parrotta[2], Annalaura Torella[4], Domenico Palumbo[5,6], Pierangelo Veltri[2], Antonella De Angelis[3], Liberato Berrino[3], Francesco Rossi[3], Alessandro Weisz[5], Marcello Rota[7], Konrad Urbanek[1], Bernardo Nadal-Ginard[2], Daniele Torella [1,9] & Eleonora Cianflone [2,7,9]

miRNAs modulate cardiomyocyte specification by targeting mRNAs of cell cycle regulators and acting in cardiac muscle lineage gene regulatory loops. It is unknown if or to-what-extent these miRNA/mRNA networks are operative during cardiomyocyte differentiation of adult cardiac stem/progenitor cells (CSCs). Clonally-derived mouse CSCs differentiated into contracting cardiomyocytes in vitro (iCMs). Comparison of "CSCs vs. iCMs" mRNome and microRNome showed a balanced up-regulation of CM-related mRNAs together with a down-regulation of cell cycle and DNA replication mRNAs. The down-regulation of cell cycle genes and the up-regulation of the mature myofilament genes in iCMs reached intermediate levels between those of fetal and neonatal cardiomyocytes. Cardiomyo-miRs were up-regulated in iCMs. The specific networks of miRNA/mRNAs operative in iCMs closely resembled those of adult CMs (aCMs). miR-1 and miR-499 enhanced myogenic commitment toward terminal differentiation of iCMs. In conclusions, CSC specification/differentiation into contracting iCMs follows known cardiomyo-MiR-dependent developmental cardiomyocyte differentiation trajectories and iCMs transcriptome/miRNome resembles that of CMs.

[1] Department of Experimental and Clinical Medicine, Magna Graecia University, Catanzaro, Italy. [2] Department of Medical and Surgical Sciences, Magna Graecia University, Catanzaro, Italy. [3] Department of Experimental Medicine, University of Campania "L. Vanvitelli", Naples, Italy. [4] Department of Precision Medicine, University of Campania "Luigi Vanvitelli", Naples, Italy. [5] Department of Medicine, Surgery and Dentistry 'Scuola Medica Salernitana', University of Salerno, Salerno, Italy. [6] Clinical Research and Innovation, Clinica Montevergine, Mercogliano, Italy. [7] Department of Physiology, New York Medical College, Valhalla, NY, USA. [8] These authors contributed equally: Mariangela Scalise, Fabiola Marino. [9] These authors jointly supervised this work: Daniele Torella, Eleonora Cianflone. ✉email: dtorella@unicz.it; cianflone@unicz.it

Cardiomyocyte (CM) specification, differentiation, and maturation are complex processes that require the activity of many temporally and spatially modulated gene regulatory networks[1–3]. microRNAs (miRs) play a major role in regulating these cardiac transcriptional pathways which drive CM development, maturation, and function[4–8]. The relevance of miRNAs in cardiac development and function is documented by the effect of cardiac-restricted knockout of Dicer, the gene encoding RNase III endonuclease essential for normal miR processing[9–11]. Cardiac progenitor-specific deletion of Dicer using a CRE recombinase under the control of the endogenous Nkx2.5 promoter at E8.5, led to embryonic lethality at E12.5[9,10]. Inactivation of Dicer using cTnt-Cre led to embryonic lethality at E15.5[11]. Finally, deletion of Dicer by α-Mhc-Cre caused death four days after birth by heart failure[9].

In vitro differentiation of embryonic stem cells (ESCs), as well as of induced pluripotent stem cells (iPSCs), into CMs recapitulate the CM development, differentiation, and maturation in the embryonic life[12–14]. These in vitro iPSC-based model systems have revealed different roles played by specific miRs in cardiomyocyte differentiation. miR-1, the most abundant miR in CMs, functions cooperatively with miR-133 to promote mesoderm formation and to suppress non-muscle gene expression in ESCs[15,16]. miR-199a-3p, miR-214-3p, miR-483-3p, and miR-322/-503 were shown to be relevant for mesodermal specification of cardiac progenitor cells[16,17]. miR-208a, -208b, and -499 are intronic miRs encoded by the *Myh6*, *Myh7*, and *Myh7b* genes, respectively. Because their expression depends on the expression of the respective myosin genes, they were named myo-miRs[18]. A combination of miR-1, miR-133, miR-208, and miR-499 was able to directly induce the in vitro cellular reprogramming of fibroblasts into CMs[15–21]. Combination of miR-125 b-5p, -199a-5p, -221, and -222 expression into ESC-derived CMs improved their maturation[22]. Let-7, a non-cardiac-specific miR, is required to enhance a number of functional properties relevant to CM maturation by promoting the metabolic transition from glucose to fatty acids[23]. Despite these findings, ESC- or iPSC-derived in vitro CMs are not fully differentiated and continue to reveal immature properties of embryonic/fetal CMs[12,24,25].

The adult cardiac stem/progenitor cells (CSCs/CPCs) are a small cohort of cardiac tissue-resident progenitor cells present in the postnatal mammalian heart[26–43]. CSCs can be cloned, propagated in long-term culture, and maintained in an undifferentiated self-renewing state; when grown in suspension they form spheres similar to pseudo-embryoid bodies called cardiospheres[44–46]. CSCs grown in differentiation media for endothelial (EC), smooth muscle (SMC), and cardiomyocyte (CM) lineages, they acquire phenotypic characteristics of these different cell types[44–46]. CSC transplantation as well as their in-situ activation were shown to differentiate into new myocytes, microvasculature, and fibroblasts in response to ischemic and non-ischemic myocardial damage[36–39,47–54]. Despite the above data, several genetic cell fate mapping studies failed to show a significant contribution of the endogenous CSCs to new CM formation in adulthood and after injury[55–60]. While we have challenged these results and shown that the significant limitation of the cell fate mapping approaches used in these studies precludes mapping the fate and the role of the CSCs in the in vivo CM renewal[43,48,49,53], from a miRNA/mRNA network perspective it is still unknown, to what extent the miRNA/mRNA profile of the CSC-derived CMs overlaps that of the adult CMs. It is also unknown whether the differentiation of CSCs into CMs undergoes the transcriptional switch typical of cell cycle exit during cardiac myocyte differentiation up-regulating known myo-miRs and more generally the set of more abundant miRs in adult CMs. The major objective of this study is to fill this gap while defining the level of maturation of the in vitro CSC-derived CMs.

## Results

### Adult CSCs in vitro differentiate into spontaneously contractile cardiomyocytes closely resembling fetal to neonatal cardiomyocytes.

Heart development from the embryonic cardiac mesoderm requires the spatially and temporally regulated expression of specific growth factors, such as Wingless/WNT (WNTs), Bone morphogenetic proteins (BMP), and Nodal/Activin molecules followed by the ensuing activation of their dependent gene cascades. These specific cardiac morphogens are also necessary and sufficient to specify and differentiate ESCs and CSCs into CMs in defined conditions in vitro[29,36,38].

A cardiomyogenic differentiation protocol initially developed for adult rat primary and clonal CSC-derived cardiospheres (CS)[29], similar to a protocol to generate cardiomyocytes from cardiovascular progenitors (CPC) from ESC-derived embryoid bodies (EB)[61], has been used here to efficiently differentiate mouse CSCs into functional beating CMs. 3 biological replicates of a single CSC clone ($1 \times 10^5$ cells for each replicate) were placed for 4 days in bacteriological dishes containing CSC growth medium to generate CSs[36,48]. The CSs were then switched to base differentiation medium (StemPro34) containing BMP-4 (10 ng/ml), Activin A (50 ng/ml), basic fibroblast growth factor (β-FGF) (10 ng/ml), WNT-11 (150 ng/ml) and WNT-5a (150 ng/ml) from day 4–8 with complete media replaced every 48 hours (see Fig. 1a–c). Differentiating CSs were then transferred to laminin-coated dishes in StemPro34 medium with Dickkopf-related protein (DKK-1) (150 ng/ml) added every 48 h together with 50% fresh medium replacement for up to 14 days to inhibit the canonical Wnt/β-catenin pathway, an inhibition shown to be required for cardiomyocyte differentiation during heart development[36–41]. The differentiating CSCs start spontaneous contractile activity and rhythmic beating at day 10 (i.e., two days after the start of DKK-1 administration) (Supplementary Movie 1).

In order to evaluate the structural characteristics of these CSCs-derived beating cells, immunocytochemistry with antibodies directed against myofilament proteins, such as cardiac Troponin I (cTnI) and myosin heavy chain (MF20 antibody) was carried out. As shown in Fig. 1e, f, both sarcomeric proteins, cTNNI and MHC, were expressed in high percentages, respectively $75 \pm 11\%$ and $83 \pm 7\%$, of CSC-derived CMs (hereafter identified as CSC-induced CMs, iCMs) of beating spheres at 10 days (Fig. 1d–f). In particular, dissociated iCMs at 14 days in differentiation medium showed well-developed sarcomeric striations that more closely resembled neonatal cardiomyocytes (neoCMs isolated from 2 days old mice) which are still immature compared with that of adult terminally differentiated CMs (aCMs, isolated from 10 weeks old mice) (Fig. 1g). qPCR analysis shows that at the end of the differentiation protocol the main cardiac transcription factors, Gata-4, Nkx2.5, Mef2C, and Hand2 were robustly up-regulated with a concurrent expression of the contractile protein genes, such as cTnnt2, Myh7, and Actc1, (Fig. 1h). Importantly, to assess the level of maturation of iCMs, we compared mRNA levels of key cardiac transcription factors and myofilament proteins of iCMs with whole fetal mouse heart from 11.5 embryonic day (for simplicity abbreviated as E11.5 Heart), neoCMs, and aCMs. As previously reported[62], mRNA levels of cardiac developmental genes, such as Nkx2.5, Gata-4, and Hand2 showed the highest expression in iCMs (Fig. 2a, b). On the other hand, all myofilament protein genes in iCMs reached an overall expression level intermediate between fetal and neonatal CMs but that still is significantly lower than aCMs (Fig. 2a, b).

The above data were generated from 3 biological replicates of a single randomly picked CSC clone. To validate these results, we have characterized the myocyte commitment in vitro from two

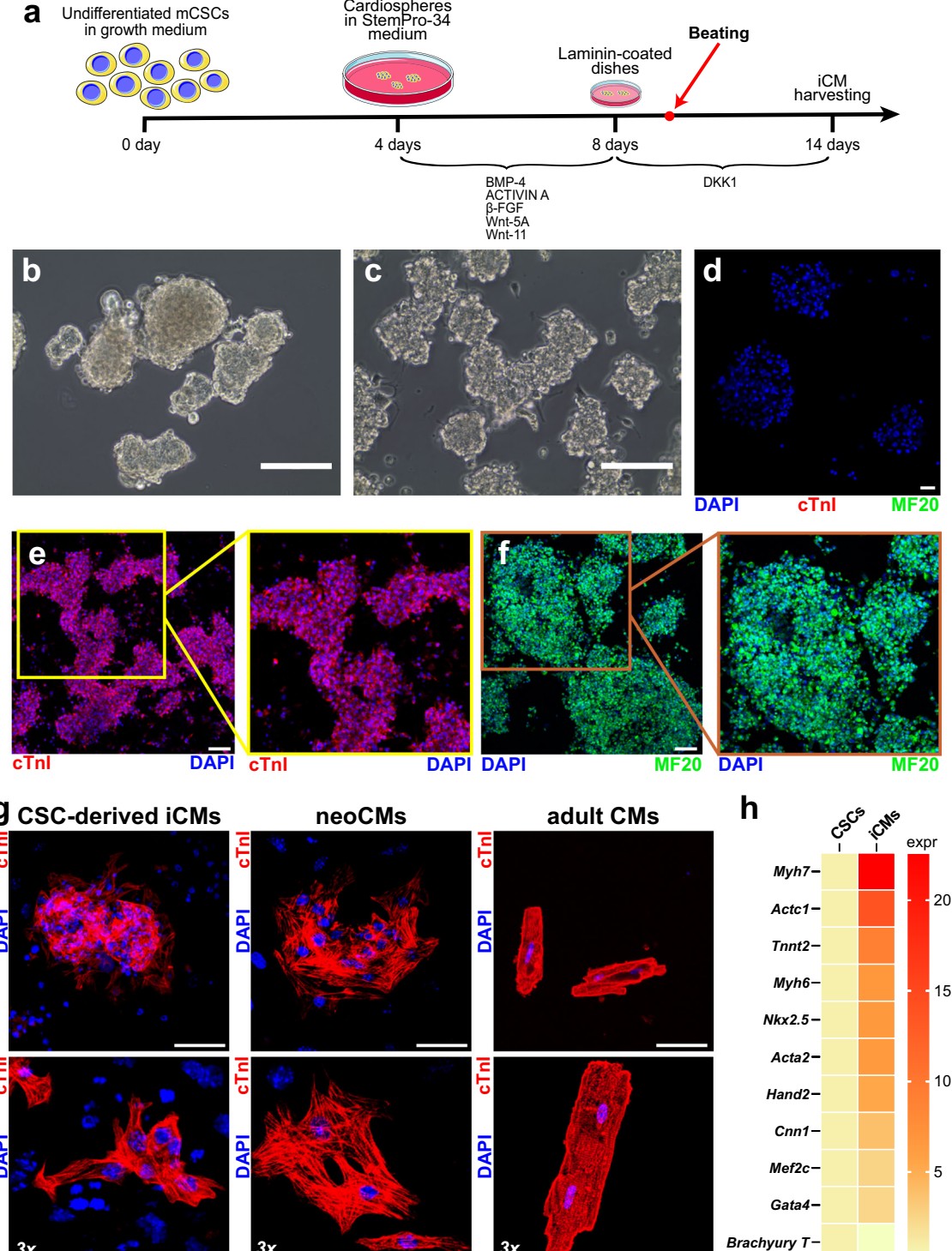

**Fig. 1 Cardiomyocyte differentiation potential of cloned CSCs in vitro. a** Schematic representation shows the differentiation protocol to derive beating cardiomyocytes from cloned CSCs in vitro. **b** Representative light microscopy image of mouse cloned CSCs-derived cardiospheres (CS) 3 days before myogenic differentiation induction. Scale bar = 200 μm. **c** Representative light microscopy image of mouse cloned CSCs-derived cardiospheres (CS) at the end of the differentiation protocol. Scale bar = 100 μm. **d** Representative confocal image of mouse cloned CSCs-derived cardiospheres (CS) 3 days before myogenic differentiation induction, showing no cTnI or MF20 expressing cells. Scale bar = 75 μm. **e, f** Representative confocal images (areas in the squares are shown at 1.4 zoom) of functional cardiomyocytes derived from CS differentiation, showing a homogenous expression of cTnI (red) and MF20 (green). Nuclei are stained by DAPI (blue). Scale bar = 75 μm. **g** Confocal microscopy examples of CSC-derived iCMs, neoCMs, and adult CMs labeled with cTnI (red) are shown at lower (upper panels) and higher magnification (lower panels). Similar morphology of CSC-derived iCMs and neoCMs is apparent. **h** Heatmap showing qPCR analysis of main contractile genes in cardiosphere-derived CSCs (*Tnnt2, Myh7, Actc1, Myh6, Acta2, Cnn1*) and cardiac transcription factors (*Mef2c, Gata-4, Nkx2.5, Hand2,* and *Brachyury T*) after myogenic differentiation. All data for each of the above panels are representative of the analysis of biological triplicates.

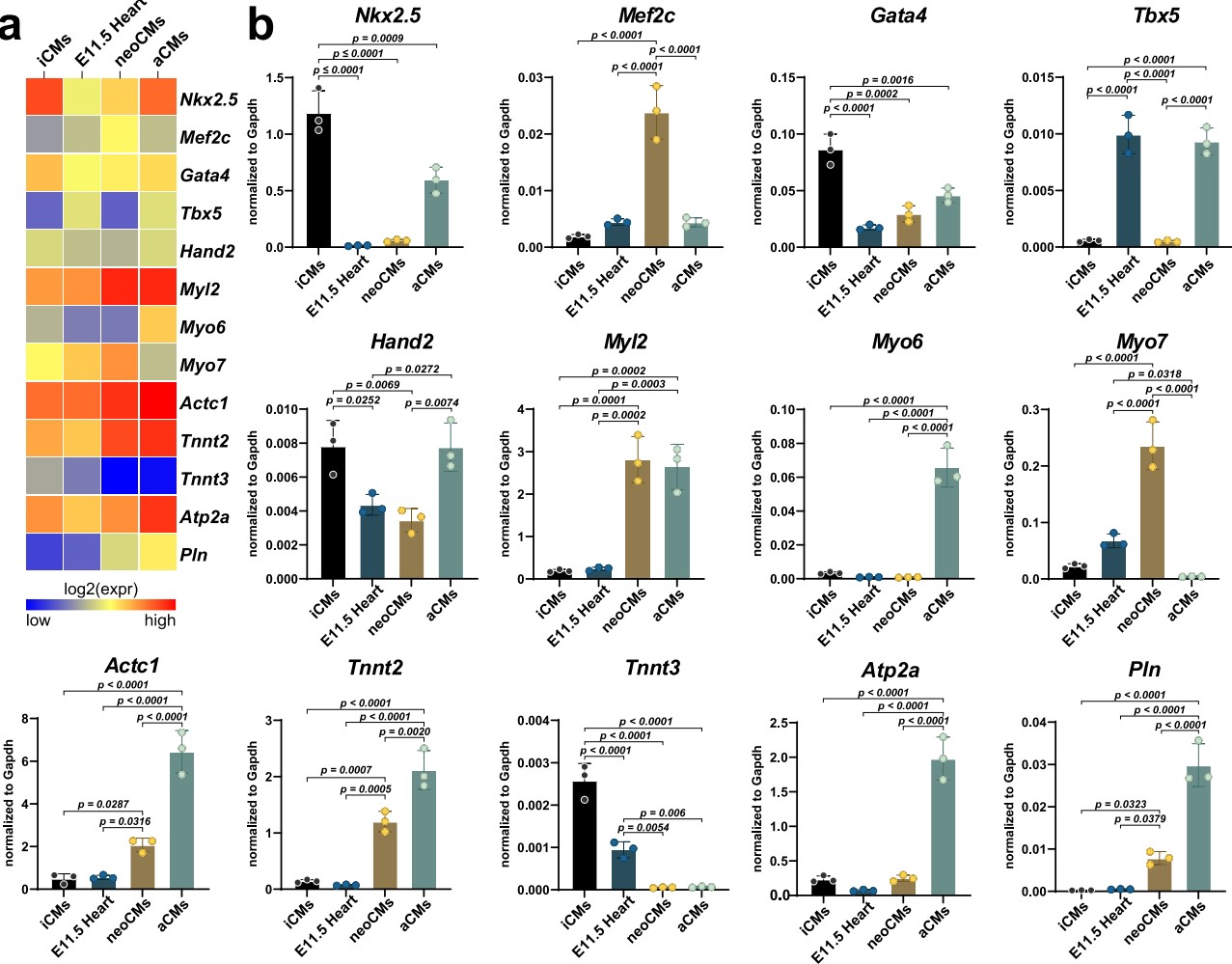

**Fig. 2 iCMs reach a biochemical myogenic differentiation level that is intermediate between fetal and neonatal CMs.** Heatmap (**a**) and bar graphs (**b**) showing the cumulative qPCR data for the expression of the main cardiac transcription factors *Gata-4, Nkx2.5, Mef2c,* and *Hand2* and the contractile genes *Tnnt2, Myl2, Myh7,* and *Actc1* in iCMs, myocardium from 11.5 embryonic day, neoCMs and aCMs. The data are expressed as mean ± S.D. of biological triplicates. Each of the biological triplicates was verified by a technical triplicate.

additional independent CSC clones. When comparing the CSC myogenic commitment in the in vitro differentiation assay of these two different clones with the one above, it is evident that they all reached similar myocyte differentiation levels when evaluated by cardiac transcription factor, contractile gene mRNA expression, and myofilament contractile protein level of expression (see Supplementary Figure 1).

Overall, the cytological, immunofluorescence, and gene expression analyses indicate that clonal adult mouse CSCs differentiate into spontaneously beating contractile cells with the structural and biochemical phenotype of immature fetal/neonatal cardiomyocytes.

**Global transcriptome of CSC-derived iCMs shows a gene expression pattern of well differentiated yet immature cardiomyocyte phenotype.** To better evaluate the transcriptional phenotype of the mouse iCMs, their global transcriptome after 14 days in differentiating media was obtained by RNASeq analysis and compared to that of their undifferentiated progenitors, the CSCs ($n = 3$ biological replicates for both iCMs and CSCs). To assess the iCMs broad cardiac muscle phenotype and the extent of their cardiomyocyte differentiation, their transcriptome was compared with that of mature adult cardiomyocytes (aCMs) isolated from 10-week old mice ($n = 2$ biological replicates).

Enrichment Analysis was used to identify the up and down-regulated gene sets grouped by the Gene Ontology resource tools[63] in the different samples.

Comparison of the differential gene expression of the "iCMs vs. CSCs" (Fig. 3a) as grouped by the Gene Ontology tools (Fig. 3b) revealed that 121 mRNAs were down-regulated while 2601 mRNAs were up-regulated in iCMs (Fig. 3a). CSCs are highly enriched with genes involved with DNA replication, cell cycle, telomere maintenance, and epigenetic regulation of gene expression which were down-regulated in iCMs (Fig. 3b). The up-regulated gene sets in iCMs are involved in myogenic cell differentiation, heart generation, BMP signaling modulating mesodermal specification and differentiation, positive regulation of metabolic processes, signal transduction, and cytokine production and secretion (Fig. 3b, d).

Conversely, the Enrichment Analysis of the "iCMs vs. aCMs" (Fig. 3c) as grouped by the Gene Ontology tools (Fig. 3d) revealed that 944 mRNAs were significantly lower while 1915 mRNAs were significantly higher in iCMs compared to aCMs (Fig. 3c). The genes having the highest expression levels in the aCMs are mainly involved with cardiac muscle contraction, cell action potential and calcium regulation (Fig. 3d). On the other hand, the genes that are more highly expressed in iCMs are involved with cellular and metabolic processes and signal transduction (Fig. 3d).

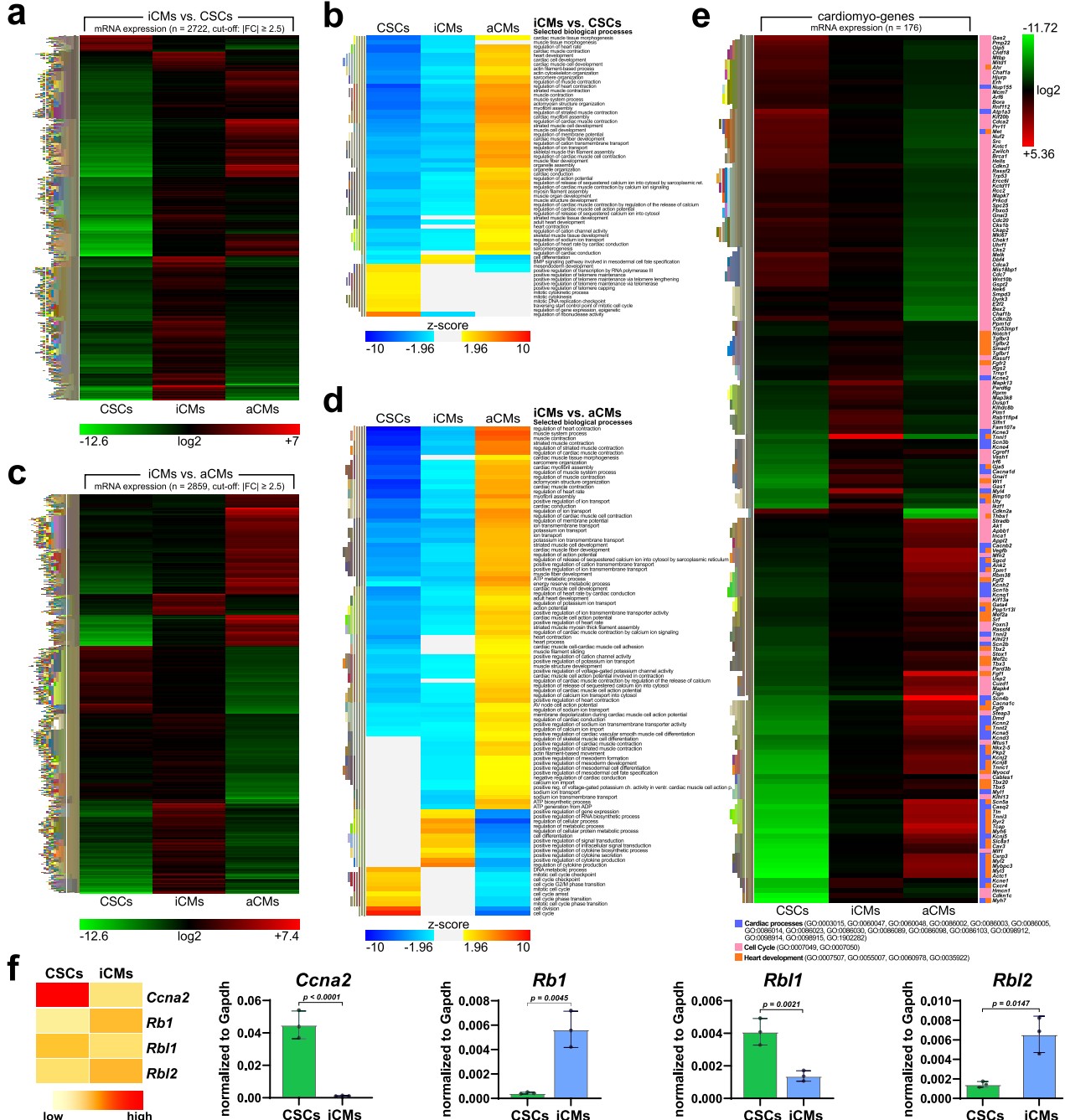

**Fig. 3 CSC-derived CMs (iCMs) show the typical gene expression profile of aCMs. a** Heatmap of RNA-sequencing profile of iCMs vs. CSCs. The comparison identified 13491 genes expressed in both cell types, with 2601 up-regulated and 121 down-regulated genes in iCMs when compared with CSCs. **b** Heatmap of RNA-sequencing profile of iCMs vs. CSCs showing their enrichment analysis for specific selected biological processes. **c** Heatmap of RNA-sequencing profile of iCMs vs. aCMs. The comparison of iCMs with freshly isolated aCMs identified 13356 expressed in both cells types, with 1915 up-regulated and 944 down-regulated genes identified in iCMs when compared with aCMs. **d** Heatmap of RNA-sequencing profile of iCMs vs. aCMs showing their enrichment analysis for specific selected biological processes. **e** Heatmap of 176 selected myogenic gene expression from RNA-sequencing analysis in the aCMs vs. CSCs. **f** Heatmap and bar graphs showing the qPCR data for the expression of mRNA for Rb pocket proteins *Rb1, Rbl1,* and *Rbl2* together with *Cyclin A2* mRNA (Cnna2) in CSCs and iCMs. The data are expressed as mean ± S.D. of biological triplicates. Each of the biological triplicate was verified by a technical triplicate. All RNASeq data are representative of biological triplicates for CSCs and iCMs and of biological duplicate for aCMs. Each of the biological replicate was verified by a technical triplicate.

Furthermore, even if with a less pronounced difference, among the mRNAs more highly expressed iCMs were those involved with cell cycle competence (Fig. 3d).

Finally, we built a list of 176 myocyte genes most expressed in the aCMs (that for simplicity we grouped under the name of 'cardiomyo-genes' to signal their high enrichment in the mature aCMs) (Supplementary Data 1). A heatmap of their expression in CSCs, iCMs and aCMs was generated (Fig. 3e). Compared to CSCs, the most up-regulated genes in iCMs were cardiac-specific transcription factors, growth factors, growth factor receptors and their main downstream signaling molecules (Mef2a, Mef2c, Gata4, Nkx2.5, Ryr2, Tbx-2, Tbx-3, Tbx-5, Tbx-20, Myocd,

Wt1, SRF, Notch1, Bmp10, Wnt10b, Cxcr4, Fgfr2, Tgfbr1, Tgfbr2, Tgfbr3, and Smad1) known to be essentials for cardiac cell commitment and differentiation during fetal development[64–66] (Fig. 3e). As expected for a population of cells committing toward a terminally differentiated lineage such as adult cardiomyocytes, iCMs show a major reduction of cell cycle genes like Cdkn2a and Cdca3 (Fig. 3e).

It is well known that the pocket protein family of tumor suppressors, and Rb specifically, are crucial for cell cycle withdrawal, maturation and maintenance of the terminally differentiated state of CMs[67,68]. In agreement with previous data[69], Rb (also known as mature Rb or Rb1) mRNA is scant or undetectable in E11.5 Hearts, but it is up-regulated in neoCMs, and it becomes the predominant pocket protein in adult CMs (Supplementary Figure 1). In contrast, p107 (also known as Rb like 1 or Rbl1) mRNA expression is reciprocal to that of Rb being highest in E11.5 Hearts and lowest in the aCMs (Supplementary Figure 1). p130 (also known as Rb like 2 or Rbl2) mRNA expression peaks in neoCMs and is subsequently down-regulated becoming very low in aCMs (Supplementary Fig. 1). This Rb pocket protein transition is necessary for aCM terminal differentiation[67,68]. Of note, iCMs significantly down-regulate cyclin A2 mRNA (Cnna2) compared to CSCs while Rb1 is up-regulated and Rbl1 is down-regulated (Fig. 3f). Accordingly, Rbl2 is up-regulated in iCMs compared to CSCs (Fig. 3f). The pattern of pocket protein gene expression further suggests that the differentiation and commitment of CSC-derived iCMs regulates the cell cycle machinery in a manner similar to the one occurring in the transition from proliferating cardiomyocytes to adult terminally differentiated CMs which occurs in the early post-natal period[67,68].

The adult CM-specific gene signature set, the myogenes, and particularly the sarcomere-related genes (i.e. Actc1, Ttn, Tnnt2, Myh6, Myh7, Myh11, Myo18b, Myl2) were up-regulated in iCMs vs. the CSCs and further up-regulated in the aCMs (Fig. 3e). Considering the immature state of iCMs[36], it is not surprising that the adult CM-specific gene signature it is not as highly expressed in the iCMs as in the aCMs (Fig. 3e). Also, because in the iCM population not all the cells are differentiated to the same degree (see above), the myogene set is diluted while the cell cycle genes are not totally down-regulated in iCMs compared to aCMs (Fig. 3e).

Overall, these data show that CSCs, when properly induced in vitro, are robustly cardiomyogenic through the activation of the entire gene network characteristic of the adult cardiomyocyte phenotype. Nevertheless, the in vitro CSC-derived CMs, even though spontaneously contracting, still have an immature cardiac muscle phenotype, closely resembling fetal/neonatal CMs which most are likely terminally withdrawn from the cell cycle a fraction of cells maintain a limited proliferative potential while lacking the levels of expression of the sarcomeric genes typical of the adult terminally differentiated phenotype.

**miRNome of CSC-derived CMs recapitulates miRNome of adult cardiomyocytes**. MiRNAs, functioning mainly as tuners and providers of the robustness of mRNA expression[70], significantly contribute to cardiomyocyte specification/differentiation and heart development. We assessed the miRNA expression profile obtained by RNASeq from the same samples used for the mRNA transcriptome analysis. To identify miRNA sequences, the reads were mapped to the mouse genome and aligned to miRBase. Based on the high-throughput sequencing results, we used a hierarchical clustering algorithm to analyze differentially expressed miRNAs in the respective biological replicates of the three cell populations: CSCs, iCMs, and aCMs. While consistent expression patterns are observed in the miRNA heatmaps intra-group, several differences

among the three biological samples were evident (Fig. 4a). To characterize these differences we performed a k-means clustering which separated miRNAs into 3 cluster groups (Fig. 4b). Cluster 1 is the group of miRNAs whose expression is significantly higher in aCMs compared to the CSCs and iCMs (Fig. 4b). Panther Classification revealed that this miRNA Cluster 1 consists of a group of miRs (listed in Supplementary Data 2) that putatively target mRNAs involved with cardiac ventricle formation and cardiac chamber formation. Cluster 2 is a group of miRNAs (see Supplementary Data 3) that are significantly higher expressed in CSCs compared to the iCMs and aCMs (Fig. 4b) and that putatively target mRNAs involved with cell proliferation. Finally, the group organized as cluster 3 (Supplementary Data 4) shows miRNAs up-regulated in iCMs compared to CSCs and aCMs (Fig. 4b) that putatively target mRNAs involved with positive stem cell population maintenance. A Principal Component Analysis (PCA) of global miRNA expression in the three cell populations locate the miRNA profile of CSCs and aCMs at opposite poles while iCMs occupy an intermediate position as expected for a cell intermediate between a progenitor and its mature progeny (Fig. 4c). The same difference among the three cell populations were also reproduced in a distance heatmap (Fig. 4d).

Volcano plots of pairwise comparison of miRNA expression between cell populations (Supplementary Fig. 2a–c) were generated. In the iCMs vs. CSCs comparison, when considering as significant a fold change of [1.5], out of a total of 1652 miRNAs, 184 were down-regulated and 283 were up-regulated in the iCMs (Fig. 4e and Supplementary Fig. 2a, Supplementary Data 5). When considering the respective mRNA targets, these down-regulated and up-regulated miRNAs in the iCMs are related to biological processes involved with cell cycle process, intracellular protein transport, RNA processing, cell division and telomere maintenance for the up-regulated miRNAs and cell-cell adhesion, regulation of membrane potential, muscle system process, potassium ion transport and heart contraction for the down-regulated miRNAs (Fig. 4f)[71–75]. On the other hand, in "iCMs vs. aCMs" comparison, out of a total of 1652 miRNAs, 175 were down-regulated and 271 were up-regulated in iCMs (Fig. 4g and Supplementary Fig. 2b, Supplementary Data 6). These differentially regulated miRNAs are involved with cation transport, inorganic ion transmembrane transport, regulation of membrane potential, $K^+$ ion transmembrane transport and action potential for the up-regulated miRNAs and with heart tube morphogenesis, regulation of cell population proliferation, regulation of cell cycle, angiogenesis, and cell cycle phase transition for the down-regulated miRNAs (Fig. 4h)[76–79].

Myogenic miRNAs describe a class of tissue-specific and developmental stage-specific cardiac miRNAs that regulate the gene expression in cardiac muscle[80,81]. Several studies have provided evidence that myogenic miRNAs play an important role in the differentiation and proliferation of muscle cells, mainly controlling cell proliferation/apoptosis, epigenetic remodeling, ion channel regulation, in addiction to the regulation of the α-MHC to β-MHC switch and in the control of the myofiber gene program identity[80,81]. Here, the miRNAs known for their involvement with myogenesis and myogenic differentiation were combined with those most expressed in the aCMs and identified as cardiomyo-miRs (Fig. 4i and Supplementary Figure 2c). Their genomic distribution is shown on the Circos plot in Supplementary Fig. 2d. In particular, cardiomyo-miRs, miR-1a 3p, miR-133a-1, miR-133a-2, miR-204, miR-335, miR-486, miR-490, and miR-499, highly expressed in aCMs (Fig. 4i) were all also up-regulated in the iCMs (Fig. 4i). Of particular interest, the levels of expression of the miR-208a and miR-208b, which are intronic to and encoded by Myh6 and Myh7 (myo-miRs), respectively, up-regulate similarly to the corresponding myosin genes in iCMs.

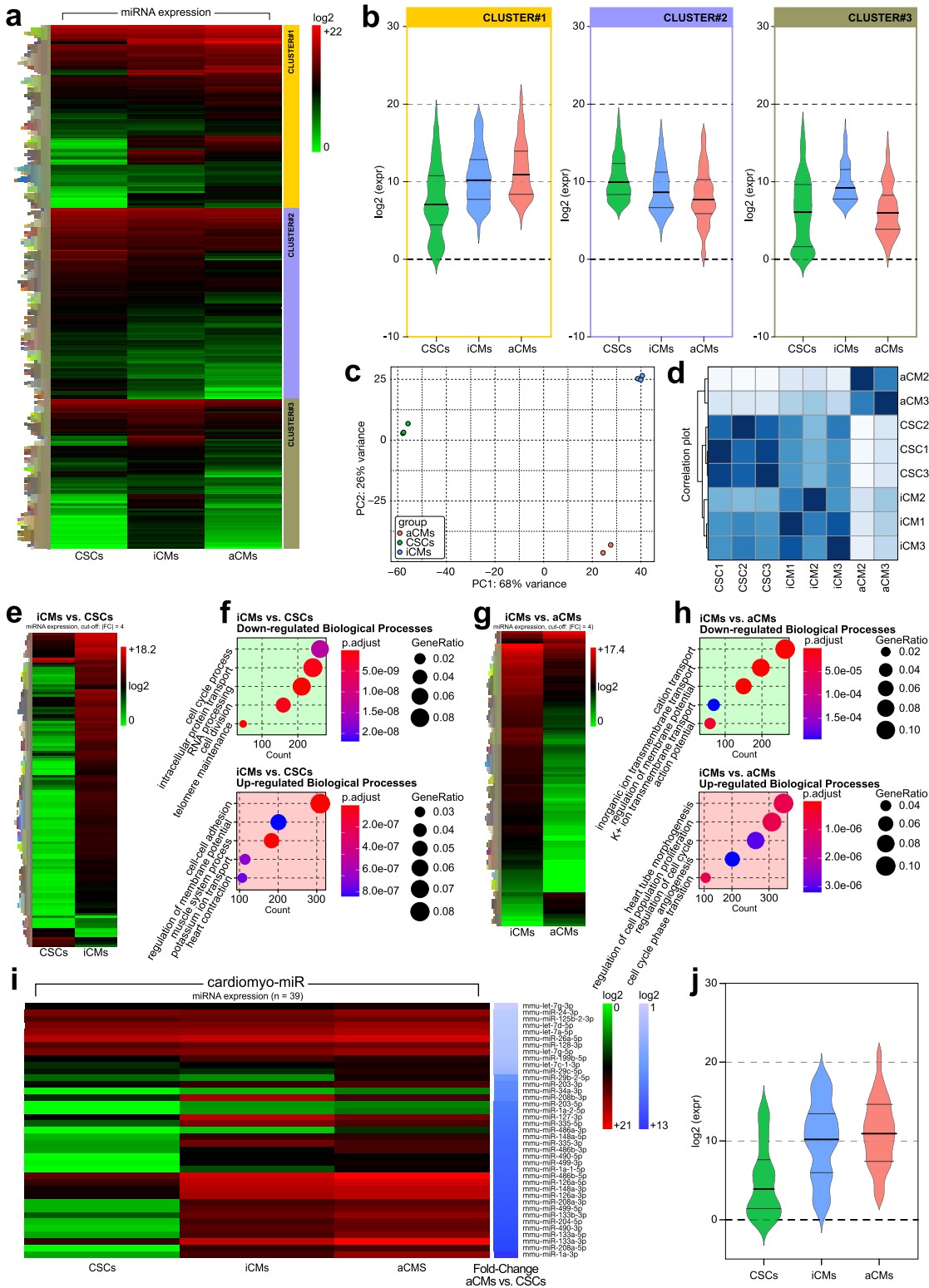

Nevertheless, these miRs up-regulated in the iCMs showed expression levels significantly lower than in the aCMs (Fig. 4i, j) as visualized in the Violin plot (Fig. 4j). In contrast to the cardiomyo-miRs, the miRNAs involved in positively regulating cell proliferation and self-renewal, such as miR-183, miR-182, miR-96, miR-298, miR-296-3p, and miR-296-5p, were highly expressed in CSCs in the comparison to aCMs (Fig. 4i).

Overall, these data suggest that cardiomyocytes derived from CSCs up-regulate the known cardiomyo-miRs, which further

confirms the bona fide cardiomyogenic phenotype of these cells. Yet, in consonance with the immature phenotype of the iCMs, most of these miRs have an expression level in these cells that is still significantly lower than in aCMs.

**Identification of miRNA/mRNA interactions and regulatory networks potentially modulating CSC specification and differentiation into cardiomyocytes.** The identification of putative

**Fig. 4 miRNASeq Expression of miRNAs associated with cardiac differentiation. a** Heatmap shows miRNA expression profile obtained by RNA-Sequencing of CSCs, iCMs, and aCMs. The average expression level of miRNAs in CSCs, iCMs, and aCMs with a number of reads greater than a mean of 5 in at least one cell population is shown. **b** Violin plots showing the log2 normalized expression read count for k-means clustering of three groups of miRNAs (clusters). Each cluster includes a group of miRNAs that have the same level of expression in the three cell populations. **c** 2D-PCA based on the miRNA expression level applied to all the replicates of the three cell populations. **d** The correlation plot shows differences between iCMs, aCMs, and CSCs in each replicate. Higher intensity represents lower variance between the cell populations. **e** Heatmap shows differential miRNA expression profile obtained by iCMs vs. CSCs comparison. **f** Dot plots GO enrichment analysis of differentially regulated miRNAs in iCMs vs. CSCs comparison. **g** Heatmap shows differential miRNA expression profile obtained by iCMs vs. aCMs comparison. **h** Dot plots GO enrichment analysis of differentially regulated miRNAs in iCMs vs. aCMs comparison. **i** Heatmap of the cardiomyo-miR expression from RNASeq analysis in the aCMs vs. CSCs comparison (with accompanying relative expression of the same miRNAs in iCMs). **j** Violin plot showing the log2 normalized expression read count of the most expressed myo-miRs in aCMs compared to iCMs and CSCs samples. All RNASeq data are representative of biological triplicates for CSCs and iCMs and of biological duplicate for aCMs. Each of the biological replicate was verified by a technical triplicate.

gene targets of each miRNA is an important first step to start elucidating its potential functions. miRNAs usually act by either inducing degradation or repressing translation of their target mRNAs[82,83]. Even though individual miRNAs are associated with specific cell functions, it is clear that the miRNome simultaneously at any given time acts over multiple mRNAs encoding proteins with similar, opposed or intertwined roles in their respective gene networks[81]. Therefore, to identify putative mRNA targets of the differentially expressed miRNAs, we performed an integrated bioinformatics analysis of the differentially expressed miRNA and mRNA in the CSCs, iCMs, and aCMs. We restricted the analysis to the validated and putative/predicted mRNA targets involved in cardiomyocyte differentiation, cell cycle, heart development, and stem cell differentiation and maintenance (Figs. 5, 6 and Supplementary Figs. 3–5).

Independent networks containing only known and predicted miRNA and their relative mRNA targets putatively or known to be involved in the process of cardiomyogenic specification and differentiation of CSCs were identified. Interestingly, the molecular networks built from the miRNA/mRNA targets comparison of iCMs vs. CSCs closely resembled the same networks constructed from the comparison of aCMs vs. CSCs (Figs. 5 and 6). In particular, the cardiomyocyte differentiation miRNA/mRNA network built on the down-regulated and up-regulated miRNAs (and respectively up-regulated and down-regulated mRNAs) was ~80% similar in its entire dataset in both, the "iCMs vs. CSCs" and "aCMs vs. CSCs" comparisons (Fig. 5 and Supplementary Data 7–8). Similarly, the miRNA/mRNA network describing the cell cycle process was similar for ~70% in its entire dataset in the "iCMs vs. CSCs" and "aCMs vs. CSCs" comparisons (Fig. 6 and Supplementary Data 9–10). As pertinent examples, miR-125b-1-3p, miR-344d-3p, miR-206-3p were down-regulated while their respective targets Nkx2.5, Tbx5, Hand2 mRNAs were up-regulated in "iCMs vs. CSCs" as well as in "aCMs vs. CSCs" comparisons (Fig. 5 and Supplementary Data 7–8). In contrast, genes involved in stem cells maintenance, self-renewal and development such as Foxp1, Cited2, Lrp6, Pdgfrα, Isl1 were found down-regulated when specific miRs such as miR-1a-3p, miR-499-5p, miR-34a-5p, and miR-335-5p were up-regulated in "iCMs vs. CSCs" as well as in "aCMs vs. CSCs" comparisons (Fig. 5 and Supplementary Data 7–8). Despite the qualitative similarities and the clear progression of CSCs toward myogenic commitment, the levels of the specific miRNAs and their putative mRNA targets in iCMs remain different from aCMs (Figs. 5 and 6).

The miRNA/mRNA network involved in the cell cycle control highlighted the down-regulation of cyclin and cyclin-dependent kinases, such as Ccnd1, Cdk6, Cdk14 and transcription factors such as E2f1, E2f3, E2f5 when miR-1a-3p, miR-499-5p, miR-34a-5p were up-regulated in both the "iCMs vs. CSCs" as well as in the "aCMs vs. CSCs" comparisons (Fig. 6 and Supplementary Data 9–10). Conversely, the down-regulation of miR-96-5p and

miR-34c-5p (as well as miR-34b-5p) was associated with up-regulation of genes involved in stem cells differentiation and cell cycle arrest such as Foxo4 and Notch1 (Fig. 6 and Supplementary Data 9–10). The miRNA/mRNA networks describing heart development, and stem cell differentiation and maintenance showed a similar trend as the above-mentioned networks (see Supplementary Figure 3–5).

Overall, despite that direct miRNA/mRNA interactions were not validated experimentally, the bioinformatics predictions outlined above supports the notion that CSC differentiation into iCMs activates cardiomyocyte-specific and cell cycle arrest gene networks similar to those orchestrating cardiac development and maturation.

**miR-1 and miR-499 enhanced myogenic commitment of CSC-derived iCMs in vitro.** The above-described miRNA/mRNA networks show that miRNA profile comparison between "iCMs vs. CSCs" resembles that between"aCMs vs. CSCs". To further validate these comparisons we analyzed by qPCR several of the up-regulated miRNAs identified by the miRNA/mRNA networks in both"iCMs vs. CSCs" and 'aCMs vs. CSCs' comparisons (Fig. 7a). miR-1 and miR-499 were the ones to show the highest expression in aCMs and are also highly expressed in the iCMs albeit at a significantly lower level than in the aCMs (Fig. 7a). Previous data has shown that these two miRs can foster the myogenic differentiations of both ESCs and cardiovascular progenitors[17,84]. Thus, we tested whether the transduction of these two miRNAs, either separately or in combination, would further enhance the myogenic commitment of CSC-derived iCMs.

CSCs were transduced with lentiviral vectors encoding for miR-1, miR-499, miR-1+ miR-499, or a scrambled miR. 72 h after the miRNA transductions, miR-1, and miR-499 levels were several folds higher when compared to the scrambled miRNA transduction (Fig. 7b). 48 h after efficient transduction with miR-1 and/or miR-499, CSC underwent the myogenic differentiation assay as shown above (Fig. 1a). 14 days later, direct targets of these two miRNAs involved with the cell cycle, such as Cyclin D2 and Cdk6 for miR-1 and Ccna2 for miR-499 were down-regulated in iCMs transduced with the respective miRs (Fig. 7c). These data show that both, miR-1 and miR-499, negatively regulate the cell cycle gene network, which is typical of the myogenic terminal differentiation of cardiac muscle cells. Interestingly, iCMs transduced with miR-1 and/or miR-499 show a significantly higher expression of Nkx2.5, Gata-4, Mef2c, Tbx5, Hand2, Myo6, Myo7, Tnnt3, Actc1, and Atp2a when compared to scrambled miRNA transduction (Fig. 7d, e). In particular, iCMs transduced only with miR-1 up-regulated several genes such as Nkx2.5, Mef2c, Gata4, Atp2a. Transduction only with miR-499 positively regulated the expression of Tnnt3 and Actc1. Interestingly, co-transduction with miR-1 and miR-499 further

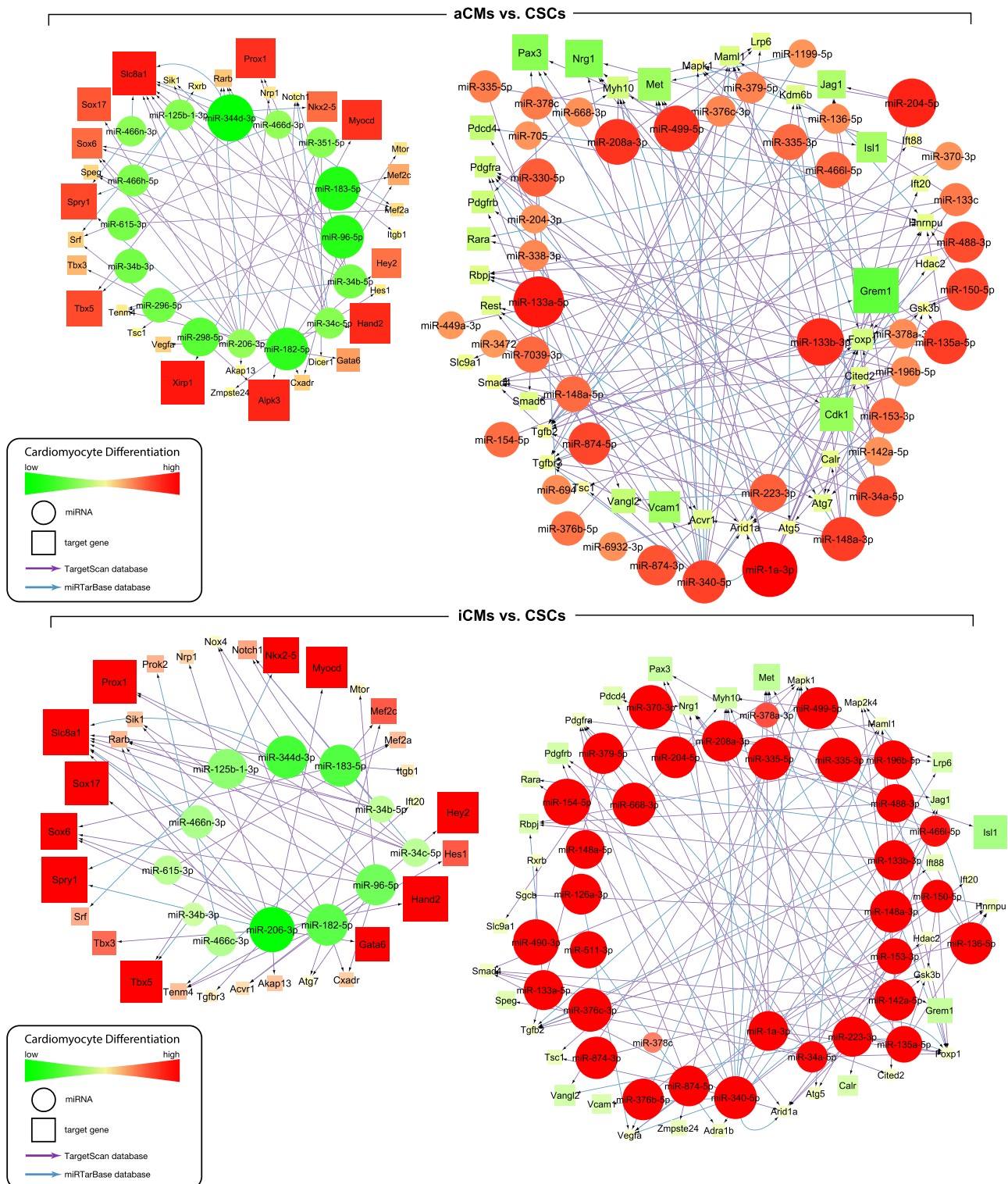

**Fig. 5 miRNA/mRNA networks describing the process of cardiomyocyte differentiation.** The networks were built starting from the down-regulated (left) and up-regulated (right) miRNAs in the "iCMs vs. CSCs" (upper panel) and "aCMs vs. CSCs" (bottom panel) comparisons, respectively. The networks were developed inserting each miRNA/mRNA target that was effectively up-regulated for the down-regulated miRNAs and each miRNA/mRNA target that was effectively down-regulated for the up-regulated miRs. Differentially expressed miRNAs were used as starting nodes to generate the interaction network of miRNAs and their targets. Each link connects the source (miRNA, circle) and its targets (mRNA, squares). The colors represent changes in expression (green as down-regulated, red as up-regulated). Subsequentially functional analysis of miRNA/mRNA networks was done for selected biological processes. All RNASeq data are representative of biological triplicates for CSCs and iCMs and of biological duplicate for aCMs. Each of the biological replicate was verified by a technical triplicate.

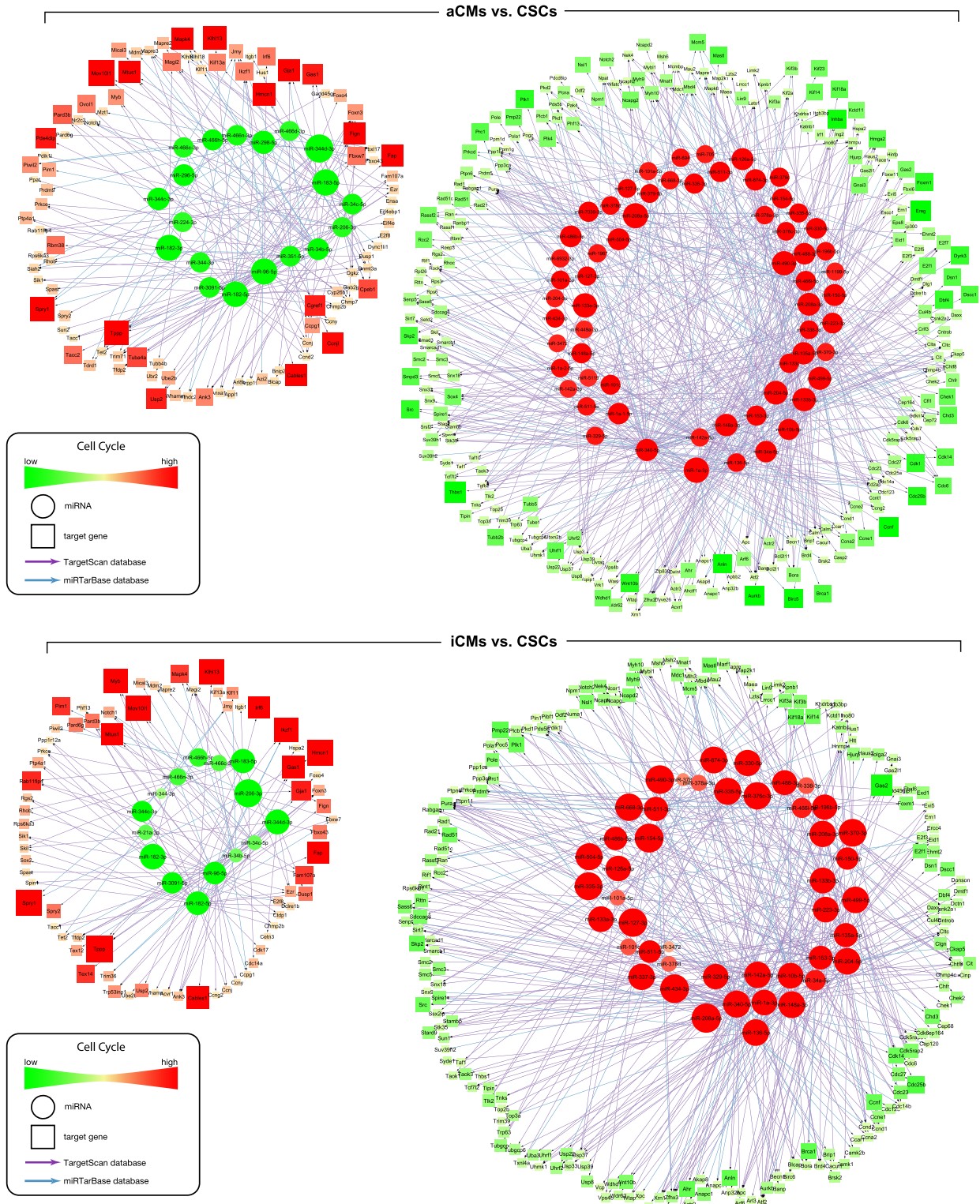

**Fig. 6 miRNA/mRNA networks describing the process of cell cycle.** The networks were built starting from the down-regulated (left) and up-regulated (right) miRNAs in the "aCMs vs. CSCs" (upper panel) and "iCMs vs. CSCs" (bottom panel) comparisons, respectively. The networks were developed inserting each microRNA mRNA target that was effectively up-regulated for the down-regulated miRNAs and each miRNA/mRNA target that was effectively down-regulated for the up-regulated miRNAs. Differentially expressed miRNAs were used as starting nodes to generate the interaction network of miRNAs and their targets. Each link connects the source (miRNA, circle) and its targets (mRNA, squares). The colors represent changes in expression (green as down-regulated, red as up-regulated). Subsequentially functional analysis of miRNA/mRNA networks was done for selected biological processes. All RNASeq data are representative of biological triplicates for CSCs and iCMs and of biological duplicate for aCMs. Each of the biological replicate was verified by a technical triplicate.

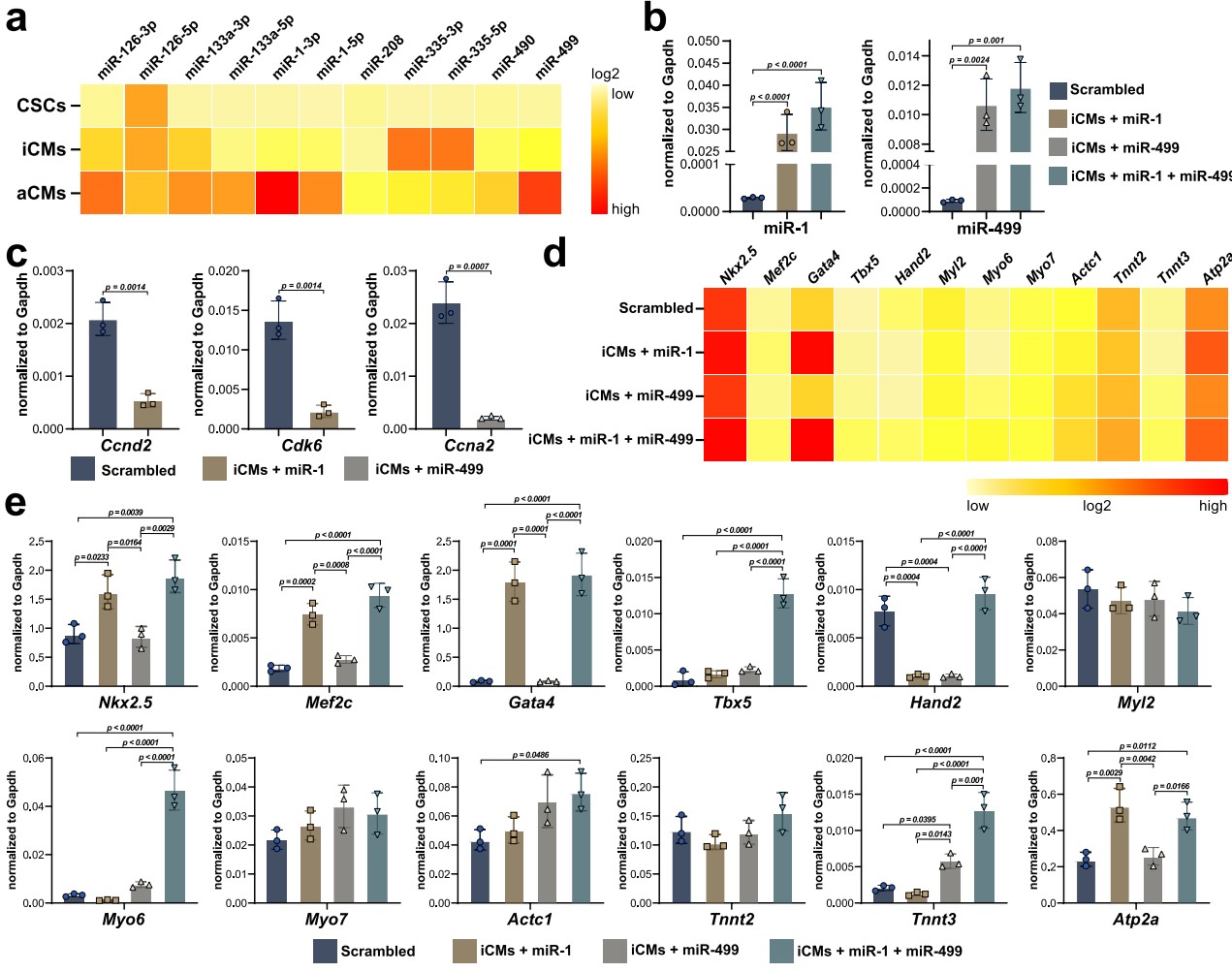

**Fig. 7 miR-1 and miR-499 enhanced myogenic commitment of CSC-derived iCMs in vitro. a** Heatmap showing the up-regulation of several miRNAs in aCMs. **b** Bar graphs showing the expression of miR-1 and miR-499 following overexpression in iCMs. **c** Bar graphs showing respectively, the expression of *Cyclin D2* and *Cdk6*, direct targets for miR-1, and the expression of Ccna2, direct target for miR-499 in iCMs transducted with the respective miRNAs. **d**, **e** Bar graphs and heatmap showing qPCR data for the expression of the main cardiac transcription factors *Nkx2.5, Gata4, Mef2c, Tbx5, Hand2, Myl2, Myo6, Myo7, Tnnt2, Tnnt3, Actc1,* and *Atp2a* in iCMs after transduction of iCMs with scrambled miR, miR-1, miR-499, and miR-1 + miR-499. The data are expressed as mean ± S.D. of biological triplicates. Each of the biological triplicate was verified by a technical triplicate.

promoted the expression of Tbx5, Hand2, Myo6, and Tnnt3, while the expression of Tnnt2, Myl2, Myo7, and Atp2a was not further increased when compared with the transduction with only a single miRNA. The expression of Myl2, Myo7, and Tnnt2 was not affected by neither single miRs nor by their combination. From these data, it is however not possible to determine whether the enhanced expression of this set of myocardial-specific genes upon the transduction of miR-1, miR-499, or both is just the indirect consequence of suppression of the cell cycle genes and/or the down-regulation of unknown negative regulator/suppressor mRNAs dampening the expression of muscle-specific mRNAs in the early stages of cardiomyogenic differentiation (Fig. 7b, d–e).

Overall, these data support the conclusion that the miRNA/mRNA network built by the bioinformatic analysis identified miRNAs involved in myocyte commitment and terminal differentiation as indeed miR-1 and miR-499 were able to enhance the myogenic commitment of CSC-derived iCMs toward the terminal differentiation and cell cycle withdrawal typical of adult CMs.

## Discussion

The principal findings of this study are as follows: (i) mouse CSC in vitro robustly differentiate into beating cardiomyocytes

(iCMs); (ii) CSC-derived iCMs transcriptome and miRnome profiles recapitulate that of adult CMs even though mRNA and miRNA levels reflect an immature phenotype closer to fetal/neonatal CMs; (iii) CSCs differentiation into CMs is characterized by the switching on of the miRNA/mRNA networks operative in the adult CMs; (iv) two miRs from these miRNA/mRNA networks, miR-1 and miR-499, individually and together enhance myogenic commitment and terminal differentiation of CSC-derived iCMs.

From embryonic to adult life, mammalian cardiomyocytes (CMs), generated from cardiac progenitors within the mesodermal layer, undergo a differentiation and maturation process characterized, among other traits, by the permanent exit from the cell cycle, development of sarcomeric structures, $Ca^{2+}$ storage and release system, binucleation and increased metabolic demand. This process is transcriptionally regulated by specific myogenic mRNAs which expression is modulated by a family of miRNAs[4]. During embryonic development, Wnt, BMP, and Nodal/Activin pathways coordinate various transcriptional and growth factors to promote cardiac progenitor cells proliferation and differentiation. BMPs are members of the transforming growth factor-beta (TGF-β) super-family that play an essential role in most of the morphogenetic processes during

development[85]. Mesodermal induction and cardiac differentiation require BMP signaling[86,87]. The activation of Wnt/β-catenin signaling promotes cardiac differentiation, but during gastrulation inhibits heart formation[88]. Early in mammalian cardiac myogenesis, Wnt/β-catenin signaling activation is indispensable for cardiac differentiation in P19 embryonic cells[89] and highlight the biphasic role of Wnt/β-catenin signaling in cardiomyocyte differentiation: activation is required to commit mesenchymal cells to the cardiac lineage; down-regulation of β-catenin is needed for cardiomyocyte differentiation at later stages. These important molecular pathways have been used in vitro to direct the differentiation of adult CSCs into CMs[36,90–92]. Here we further show that a refined protocol of sequential and step-by-step administration of BMP-4, Activin A, β-FGF, and DKK-1 to clonal CSCs increase the expression of myogenic genes and the number of Troponin or Myosin positive cells, generating spontaneously beating cardiomyocytes in vitro (iCMs).

Transcriptome analysis is a powerful genomic tool that facilitates the identification of whole gene expression networks and regulatory mechanisms[93]. In this study, whole gene expression profiling by mRNASeq analysis together with different bioinformatic tools of CSCs and CSC-derived iCMs revealed a molecular signature in iCMs that closely resembles that of adult terminally differentiated CMs (aCMs). Indeed, the most relevant enriched pathways and functions describing the comparisons of CSCs with either iCMs or aCMs were similar and related to cell cycle progression, cardiac function and maturation.

Also, genome-wide small RNA sequencing identified candidate miRNAs involved with CSCs differentiation into iCMs. Comparison of the miRNome profiles of CSCs with those of iCMs and aCMs identified miRNAs that potentially play a role in the development of the CM phenotype. Interestingly, iCMs express a miRNA profile similar to the miRs set typical of adult CMs.

Despite the robust activation of a proper myocyte differentiation molecular and cell functional program, the quantitative expression of myogenic miRNAs and related mRNAs in the iCMs is intermediate between E11.5 fetal and neonatal cardiomyocytes[36]. In part, this immature phenotype of the iCMs might be due to the fact that we compared the bulk population of iCMs deriving from CSCs, which is highly heterogeneous in terms of the differentiation stage of the cells, with pure clonal CSCs and pure isolated aCMs. Thus, even though the differentiation protocol is robust and highly efficient we didn't compare a pure population of iCMs with pure CSCs and aCMs. Single-cell mRNA and miR profiling will be required to determine the stage of differentiation the iCMs can reach in vitro.

The bioinformatics tools built miRNA/mRNA networks operative in both, iCMs and aCMs, strongly suggesting that CSC-derived iCMs are generated through well-orchestrated miRNA/mRNA developmental networks from stemness/progenitor to CM-specific which are also operative in the aCMs and highlight potential targets to improve iCM maturation. Indeed, miR-1 and miR-499, already known for their role in the differentiation of ESCs and adult cardiac progenitors into CMs[17] when up-regulated in the iCMs improved the maturation of these cells towards the typical terminal differentiation of aCMs. miR-1 and miR-499, independently and together, further reduced the expression of positive cell cycle regulatory genes, such as CyclinA2 and Cyclin D, while increasing the expression of negative cell cycle regulatory genes, such as Rb1, which are known to be crucial for CM terminal differentiation[69,70,93]. Furthermore, miR-1 and miR-499 further increased the expression of sarcomere genes in iCMs.

Overall, this study focused on the efficiency of and the molecular mechanisms underlying the in vitro myogenic differentiation of cells derived from individual clones of CSCs obtained from adult hearts. The data presented indisputably show the stable myogenic potential of these CSCs under the conditions

tested. Moreover, the mRNA and miRNA changes, underlying the transition from replicating CSCs to terminally differentiated but still immature cardiomyocytes, qualitatively closely resemble those observed in the mature cardiomyocytes isolated from the adult heart. Therefore, these data further reinforce the conclusion that the CSCs are *bona fide* cardiac myogenic stem/progenitor cells, and their phenotype is consistent with previous reports of their role in vivo in adult cardiomyocyte homeostasis and myocardial repair and regeneration[36,41,48]. However, given the controversy surrounding the role of the CSCs in vivo, it is imperative to point out that the relevance of the in vitro results presented here to in vivo cardiomyocyte generation/regeneration by CSCs is just consistent with but indirect as an issue not specifically addressed here. There is a hot ongoing controversy with several scientists who have expressed a vivid skepticism about the existence of CSCs and who remain unconvinced of their role in adult cardiomyogenesis in vivo[55,58,60,94]. Indeed, recent attempts to genetically determine CSC-derived new cardiomyocyte formation using Cre-lox-based cell-fate mapping studies failed to show a significant contribution of the CSCs to new cardiomyocyte formation, let them to practically deny the existence of any adult cardiomyocyte progenitor[55,58,60,94]. That these genetic cell-fate mapping approaches fail to test the fate of the *bona fide* CSCs (the ones used in this report) has been unfortunately insufficient to resolve the controversy[48,53,95]. Consistent with the results presented here, we have previously shown that the progeny of a single CSCs isolated as described here, as opposed to the heterogenous mix of freshly isolated c-kit^pos cardiac cells[36], when exogenously transplanted in an injured heart, efficiently differentiate, and robustly replace cardiomyocytes in vivo[36,96]. Importantly, we have shown that in vivo these cardiomyocytes derived from the transplanted CSCs progressively mature over time[96]. Yet, what the field urgently needs in order to settle the controversy over the role of the CSCs in pre-and post-natal cardiac development, myocardial cell homeostasis, and myocardial repair and regeneration is an unimpeachable genetic cell-fate mapping of the CSCs which will only be possible when the proper animal genetic model becomes available[95]. In the same vein, whether the phenomenon of efficient maturation of CSC-derived cardiomyocytes in vivo depends on miR-1 and miR-499 up-regulation or whether it can be further boosted by miR-1 and miR-499 up-regulation remains to be assessed in future studies.

In conclusion, under proper in vitro culture conditions, CSCs robustly differentiate into functional beating cardiomyocytes undergoing changes in their transcriptome/miRnome which closely resembles that of adult CMs further supporting the true myogenic stem/precursor nature of these cells.

## Methods

**Cell culture.** Clonal cardiac stem cells (CSCs) were isolated from adult mouse hearts by enzymatic dissociation using gentleMACS Dissociator (Miltenyi Biotec) as previously described[36,48]. Briefly, to obtain wild type CSC clones, 6 wild-type 12-weeks-old adult C57BL/6 J mouse hearts were isolated and through negative/positive cell sorting, CD45^negCD31^negc-kit^pos cardiac cells were harvested and expanded in culture through 4 passages. At this point, ~10^3 single cells were deposited in 96-well gelatin-coated Terasaki plates by serial dilution and 14 days later, 3 clones with the fastest expansion rate were picked. These 3 clones were then expanded in CSC media (see below) for 4 passages. As we have already shown that using this isolation method, the harvested clones behave practically in an indistinguishable manner both biologically as well as transcriptionally[36], a single randomly chosen clone in biological triplicates was used to perform most of the experiments. The other two independent CSC clones were used to validate the main findings obtained with the first clone where specifically reported. Mouse CSCs clones were expanded on gelatin-coated dishes in CSCs growth medium containing of a 1:1 ratio of Neurobasal medium (Gibco, Life Technologies) and DMEM-F12Ham's (Gibco, Life Technologies) implemented with insulin-transferrin-selenium (1%, Life Technologies), epidermal growth factor (final medium concentration: 20 ng/ml, Peprotech), basal fibroblast growth factor (final medium concentration: 10 ng/ml, Peprotech) and leukemia inhibitory factor (final

medium concentration: 10 ng/ml, Miltenyi Biotec) and containing 37 mg of L-glutamine, B27 supplement (2%, Life Technologies) and N2 supplement (1%, Life Technologies) penicillin-streptomycin (1%, Life Technologies), Fungizone (0.1%, Life Technologies) and gentamicin (0.1%, Life Technologies) sterilize through a 0.22 μm pore filter into a sterile container and stored at 4 °C. The CSCs growth medium was supplemented with 10% ESQ-FBS (Life Technologies). Cells were maintained at 37 °C in ambient $O_2$ (21%) and 5% $CO_2$. Media were replenished every 48 h and cells were passaged at a 1:4 ratio.

**Cardiac differentiation protocol in vitro.** Cloned CSCs were placed in bacteriological dishes for 4 days for cardiospheres generation in the CSC growth medium. Cardiospheres were then switched to base differentiation medium consisting of StemPro®-34 SFM (a serum-free medium conditioned with StemPro®-Nutrient Supplement, Gibco, Life Technologies), Ascorbic Acid (0.5 Mm, Sigma), 1-thioglycerol ($4.5 \times 10^{-4}$ M, Sigma), L-glutamine (2 mM), Non-Essential Amino Acids (Gibco, Life Technologies) and penicillin-streptomycin (1%, Life Technologies). For specific myocyte differentiation BMP-4 (10 ng/ml, Peprotech), Activin-A (50 ng/ml first day and then 10 ng/ml, Peprotech), β-FGF (10 ng/ml, Peprotech), Wnt-11 (150 ng/ml, R&D System) and Wnt-5a (150 ng/ml, R&D System) were added to base differentiation medium from day 4 to day 8. Then, differentiating cardiospheres were pelleted and transferred to laminin (1 μg/ml) coated dishes and Dkk-1 (150 ng/ml, R&D System) was added to base differentiation medium from day 8 to day 14. Differentiated cardiospheres were either trypsinized for RNA isolation or fixed with 4% PFA.

**Mouse cardiomyocytes isolation.** Cardiomyocytes were isolated from fetal (n.3 pregnant mice), neonatal (n.3 mouse littermates with 9–10 puppies each) and adult (n.6, 12-weeks-old) C57BL/6 J mouse hearts by enzymatic dissociation. All animal experimental procedures were approved by Magna Graecia Institutional Review Boards on Animal Use and Welfare and performed according to the Guide for the Care and Use of Laboratory Animals from directive 2010/63/EU of the European Parliament. Embryonic hearts were isolated from 3 pregnant mice at the indicated time using GentleMACS Dissociator and manufacture instructions were followed. Neonatal hearts obtained from 3 different mouse littermates were placed in a plate containing MEM/Trypsin (20 μg/ml) implemented with a solution containing 0.1% DNAse I, gentamicin and fungizone (2 μl/ml), penicillin-streptomycin (1%), then cut into small pieces and gently triturated with a scissor. The plate was transferred on a stirrer at 37 °C for 10′ and then the solution was left to deposit for several seconds. The process was repeated several times. Adult hearts from 6 anesthetized mice were excised, the respective aorta cannulated and hung on a retrograde perfusion system (Langendorff method), then perfused with enzyme-containing solutions as previously described[48]. A calcium-free solution was used to digest the hearts and prepared as follows: sodium chloride at a final concentration of 126 mM; glucose at a final concentration of 22 mM; HEPES at a final concentration of 24 mM; potassium chloride at a final concentration of 4.4 mM; magnesium dichloride at a final concentration of 5 mM; creatine at a final concentration of 5 mM; taurine at a final concentration of 20 mM; sodium pyruvate at a final concentration of 5 mM and Sodium dihydrogen phosphate at a final concentration of 1 mM. To increase the cardiomyocytes yield 2,3-butanedione monoxime 10 mM was added to the solution. The solution was filtered through a 0.22 μm-pore filter into a sterile container and stored at 4 °C for up to 1 week. For heart isolation, a total amount of 50 ml of buffer was used. Briefly, the cannulated hearts were first perfused with the calcium-free solution, followed by type II collagenase digestion in presence of $Ca^{2+}$ 0.1 mM and then washed with $Ca^{2+}$ 0.1 mM solution. When perfusion was completed, the hearts were taken down from the cannula, cuts into small pieces and gently triturated with a pipette. To enrich the cell suspension with viable cardiac myocytes and to remove large or undigested tissue, preparations were filtered through a 100 μm cell strainer and then centrifuge to separate cardiomyocytes from other cardiac cell types. Viable cardiomyocytes were then allowed to sediment by gravity. The sedimentation procedure was repeated several times. The isolated rod-shaped cardiomyocytes from 2 hearts were randomly pooled to obtain biological triplicates from the 6 harvested hearts to be used for experimental purposes.

**Quantitative RT-PCR (qPCR).** RNA was extracted from CSCs, iCMs and aCM, using TRIzol Reagent (Ambion) and quantified using a Nanodrop 2000 Spectrophotometer (Thermo Fisher Scientific). Reverse transcription was performed with 0.5–1 μg of RNA using the High Capacity cDNA Kit (Applied Biosystems). Quantitative qPCR was performed using TaqMan Primer/Probe sets (Applied Biosystems) using StepOne Plus real Time PCR System (Applied Biosystems). TaqMan Gene Expression Assay was used for quantification of *Brachyury T* (ID: Mm00436877_m1), *Mef2c* (ID:Mm01340842_m1), *Gata-4* (ID:Mm00484689_m1), *Nkx2.5* (ID:Mm01309813_s1), *Tbx-5* (ID: Mm00803518_m1), *Myl2* (ID:Mm00440384), *Hand2* (ID: Mm00439247_m1), *Myh6* (ID:Mm00440359_m1), *Myh7* (ID:Mm01319006_g1), *Tnnt2* (ID: Mm00441922_m1), *Actc1* (ID: Mm01333821_m1), *Tnnt3* (ID: Mm0126886_m1), *Atp2a* (ID:Mm01201431_m1), *Pln* (ID:Mm04206541_m1), *Ccna2* (ID: Mm00438063_m1), *Rb1* (ID: Mm00485586_m1), *Rbl1* (ID: Mm01250721_m1), *Rbl2* (ID: Mm01242468_m1), *Cnn1* (ID: Mm00487032_m1) and *Acta2* (ID: Mm00725412_s1). miRNA

expression was evaluated using TaqMan miRNA assays (Applied Biosystems) for mir-1 (ID: 002222), mir-1-5p (ID: 464216_mat), mir-126-3p (ID: 000451), mir-126 (ID: 002228), mir-133a-5p (ID: 001637), mir-133-3p (ID: 002246), mir-335-5p (ID: 000546), mir-335-3p (ID: 002185), mir-208 (ID: 000511), mir-490 (ID: 001037) and mir-499 (ID: 001352). Data were processed by the ΔCt method using StepOne Software v2.3 and mRNA was normalized to the housekeeping gene, *Gapdh* (ID: Mm99999915_g1) and U6 (ID: 001973). All reactions were carried out in triplicate.

**Immunocytochemistry.** For myogenic differentiation, cardiospheres derived from cloned CSCs, were cultured on glass chamber slides for 14 days and, after fixation with 4% PFA for 20 min on ice, stained with rabbit anti-cardiac troponin I (1:100 dilution, Abcam) or mouse anti-MF20 (1:50 dilution, DSHB). Neonatal cardiomyocytes and adult cardiomyocytes were fixed with 4% PFA for 20 min on ice, stained with rabbit anti-cardiac troponin I (1:100 dilution, Abcam). Cells were incubated with anti-rabbit-488 or anti-mouse-594 secondary antibodies (1:100 dilution, Jackson Immunoresearch). Nuclei were counterstained with the DNA binding dye, 4,6-diamidino-2-phenylindole (DAPI, Sigma) at 1 μg/ml. Fluorescence was visualized and images were acquired with confocal microscopy (LEICA TCS SP8).

**RNA-seq**

*Cell samples.* Cell samples for RNA-Seq analysis were obtained from cloned CSCs, iCMs and three biological replicates to archive the analysis were performed. aCMs samples were obtained from hearts isolated from adult C57BL/6J mice and two biological replicates to archive the analysis were performed.

*RNA extraction.* RNA was extracted from CSCs, iCMs, and aCMs using TRIzol Reagent (Ambion). The extracted RNA integrity was measured with an Agilent 2100 Bioanalyzer (Agilent Technologies, Santa Clara, CA, USA), and a sample with an RNA integrity number ≥8 was considered acceptable.

*Library preparation.* The libraries were generated using depleted RNA obtained from 1 μg total RNA by TruSeq Sample Preparation RNA Kit (Illumina, Inc., San Diego, CA, USA) according to the manufacturer's protocol without further modifications. The first step involves the removal of ribosomal RNA (rRNA) using biotinylated, target-specific oligos combined with Ribo-Zero rRNA removal beads that depletes samples of both cytoplasmic and mitochondrial rRNA. Following purification, the RNA is fragmented into small pieces using divalent cations under elevated temperatures. The cleaved RNA fragments are copied into the first-strand cDNA using reverse transcriptase and random primers, followed by second-strand cDNA synthesis using DNA Polymerase I and RNase H. These cDNA fragments then have the addition of a single 'A' base and subsequent ligation of the adapter (multiple indexing adapters were ligated to the ends of the ds cDNA). The products are purified and enriched with PCR to create the final cDNA library.

*Sequencing.* All libraries were sequenced on the Illumina HiSeq 1000, generating 100 bp paired-end reads. The libraries were divided into two groups depending on how they were prepared.

*Bioinformatics analysis: mapping reads to the reference genome.* To perform RNA-seq data analysis, we used Strand NGS tools (http://www.strand-ngs.com). We set transcriptome and genome with novel splices as alignment and we use Ensembl (http://www.ensembl.org/index.html) as a transcript model. Raw data sequences (that is, reads) have been pre-processed and then been related to mus musculus genome. Parameters have been set as follows: (i) minimum percentage identity = 90%; (ii) maximum percentage gaps = 5%; (iii) maximum novel splices = 1; and (iv) minimum match length = 25. Reads with < 5 valid matches have been filtered out, finally obtaining expression values for reads (quantification phase). To detect novel genes and exons we set the parameters, in term of percentile, as follows: minimum exon length equals to 10.0; minimum intron length equals to 10.0; maximum intron length equals to 90.0; minimum gene length equals to 10.0; minimum exon RPKM equals to 50.0, where RPKM, stands for reads per kilobase per million reads, used to represent read counts over a region of interest.

Differential expressions were performed using the DESeq algorithm, available in StrandNGS (Strand Life Sciences, Bangalore, KA, India). We performed FC analysis on filtered expression values using normalized signal values that are in log-scale. We considered a |FC| cut off equals to 2.5 to identify up and down-regulated genes Heatmaps have been created using Gitools (http://www.gitools.org/) and a hierarchical clustering has been performed using Euclidean distance metric and complete (maximum) linkage clustering as a linkage method. Enrichment analysis has been performed with Gitools importing the data from the Gene Ontology databases and relating genes to biological processes, finding significantly enriched GO terms to be compared with the data set analysed[97]. Statistic values are used to select the more interesting set of genes in terms of biological processes to annotate and guide clustering analysis. To discriminate between over- and under-expressed pathways we selected to display the Z-score value, omitting biological processes having less annotated genes than 20, sampling size 10000, mean estimation and Benjamini-Hochberg (BH) multiple test correction.

## smallRNA-Seq

*Libraries preparation.* smallRNA-Seq libraries were prepared as previously described[35,98]. In details, libraries were generated from 120 ng using TruSeq Small RNA Sample Prep Kits (Illumina Inc.). Libraries were sequenced in single read mode, $1 \times 50$ cycles on HiSeq 2500 (Illumina Inc.). The quality of the sequenced data was assessed using FastQC (FastQC: A Quality Control Tool for High Throughput Sequence Data) and they were trimmed using Cutadapt[99] with default parameters. Subsequently, the filtered data were analyzed using iSMART[100]. The raw counts were normalized using DESeq2[101] (version 1.28.1) on R (version 4.0.2). The differentially expressed miRNA were filtered with a p-adjust < 0.05 and a Fold Change of |1.5| (Log2 FC | 0.6 |). Volcano plots were generated using Enhanced-Volcano library on R. Circos plots were generated using Circos (version 0.69-9)[102]. PCA plots were created using R.

*Target prediction.* mRNAs targets of differentially expressed miRNAs were detected with CyTargetLinker v4.1.0[103], an open source network tool of Cytoscape, investigating mirtarbase v8.0 and Targetscan v7.2 databases.

*Functional annotation.* Gene Ontology enrichment analyses were performed using Genome-wide annotation for Mouse (R package version 3.8.2), corrected for multiple testing p-value using the BH method. Only selected biological processes showing an adjusted *p*-value ≤0.05 were considered.

*Integrated analysis of microRNA and mRNA expression profiles.* Integrated analysis was performed on miRNAs that were significantly modulated (|FC| = 5) in CSCs vs aCMs comparison. Differentially expressed miRNAs were selected and used in Cytoscape as starting nodes (circular nodes). The interaction network of miRNAs and their targets were extended from miRTarBase (validated interactions) and TargetScan (predicted interactions) using CyTargetLinker[103]. Each link contains two connected nodes, source (miRNA) and target (mRNA). The miRNAs and mRNAs were coloured based on changes in gene expression (green as down-regulated, red as up-regulated). We next performed, a functional analysis of all the nodes of the miRNA-mRNA target networks using Gene Ontology and GO Annotations[104,105] and Panther Classification System[106]. Selected biological processes were imported as networks in Cytoscape, then merged for each of our extended miRNA-mRNA target networks.

RNASeq data are available in GEO with the accession number GSE161081.

**Lentivirus-mediated miRNA expression.** Lentivirus-vector-overexpressing miR-NAs were generated by Abmgood and contained an expression module for GFP that enabled the detection of cells with positive infection and transduction. Lentivirus-mediated miRNA expression was conducted on mouse CSCs. Cells were seeded at a density of $0.5 \times 10^5$ cells in 12-well plates and incubate at 37 °C with 5% $CO_2$ overnight. The day after, cells were infected with lentivirus at 50 multiplicity of infection (MOI) using polybrene at a concentration of 6 μg/ml. The medium was replaced with a fresh medium 24 h after infection. Infected cells were selected for the stable expression of each lentivirus vectors using puromycin (4 μg/ml) for 48 h. Quantification of miRNA was performed in two-step qPCR according to the TaqMan MicroRNA Assays protocol.

**Statistics and reproducibility.** Statistical analysis was performed with GraphPad Prism version 6.00 for Macintosh (GraphPad Software). Quantitative data are reported as mean ± SD and binary data by counts. Significance between 2 groups was determined by Student's *t*-test or paired *t*-test as appropriate. For comparison between multiple groups, ANOVA was used. A P value<0.05 was considered significant. Bonferroni post-hoc method was used to locate the differences. In these cases, the Type 1 error (α = 0.05) was corrected by the number of statistical comparisons performed. For the in vitro cell and molecular biology experiments, giving the low number of the sample, the Kruskal–Wallis test (for multiple comparisons), and the Mann group –Whitney U test (for comparison between 2 groups) were performed.

**Reporting summary.** Further information on research design is available in the Nature Research Reporting Summary linked to this article.

## Data availability

The data that support the findings of this study are available within the paper and its supplementary information files. Source data underlying figures are presented in Supplementary Data 11. RNA-Seq data that support the findings of this study have been deposited in GEO with the accession code "GSE161081". Any remaining information can be obtained from the corresponding author upon reasonable request.

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

## Acknowledgements

This research was funded by grants from the Ministry of Education, University and Research (PRIN2015 2015ZTT5KB_004, 2017NKB2N4_005), PON-AIM – 1829805-2, PON03PE00009_2-iCARE and ARS01_01226-PerMedNet (CUP: D26C18000260005) and POR Prodotti Alimentari.

## Author contributions

Conceptualization, D.T. and E.C.; methodology, D.T., K.U., A.DeA., P.V., A.W.; formal analysis, M.S., F.M., L.S., D.C., T.M., A.B., D.P., A.T.; resources, E.I.P., L.B.; writing—original draft preparation, E.C., M.S., K.U., D.T.; writing—review and editing, E.C., F.R., M.R., K.U., B.N-G., D.T.; senior authorship, D.T. & E.C. All authors have read and agreed to the published version of the paper.

## Competing interests

The authors declare no competing interests.
