## [Transparent Peer Review File · Communications Biology]

Reviewers' comments:

Reviewer #1 (Remarks to the Author):

The manuscript by Scalise et al. describes a well-defined in vitro procedure for the contractile cardiomyocyte differentiation (iCMs) from clonally-derived mouse CSCs. The protocol is similar to that described to generate cardiomyocytes from cardiovascular progenitors (CPC) from ESC-derived embryoid bodies (EB). iCM were nicely characterized and, in particular, it is described that dissociated iCMs at 14 days in differentiation medium showed a well-developed sarcomeric striations that more closely resembled neonatal cardiomyocytes (neoCMs isolated from 2 days old mice) which are still immature compared with that of adult terminally differentiated CMs (aCMs, isolated from 10 weeks old mice). RT-qPCR analysis shows that at the end of the differentiation protocol the main cardiac transcription factors, Gata-4, Nkx2.5, Mef2C and Hand2 were robustly up-regulated with concurrent expression of the contractile protein genes, such as cTnnt2, Myh7 and Actc1.

The RNASeq comparison of "CSCs vs. iCMs" mRNome and microRNome showed a balanced up-regulation of sarcomere and mitochondrial related mRNAs coupled to the a down-regulation of cell cycle and DNA replication mRNAs. Results indicated that the differentiation process is somehow incomplete not reaching the levels found in purified terminally differentiated adult cardiomyocytes (aCMs), showing intermediate levels between those of fetal and neonatal cardiomyocytes; this was confirmed by RT-qPCR analyses. In agreement, cardiomyo-miRs were clearly up-regulated in iCMs whereas those miRs positively regulating stem cell expansion and self-renewal were down-regulated. Again, the levels of regulation of key miRNAs are also lower in iCM compared with aCM. Therefore authors conclude that the specific miRNA/mRNAs networks operative in iCMs closely resembled miRNA/mRNA networks of aCMs and proposed that the difference in intensity could be associated to the heterogeneity of the iCMs population. The integrated bioinformatics analysis yield compatible miRNA/mRNA interactions supporting the notion that CSC differentiation into iCMs activates cardiomyocyte-specific and cell-cycle arrest gene networks similar to those orchestrating cardiac development and maturation.

With no doubt, although ideally the described in vitro procedure for differentiation of contractile CM has a significant margin to be improved, it is the best procedure described using adult cardiac progenitors or CSCs.

As a validation for the differentiation protocol authors engineered CSC for overexpression of miR-1, miR-499, or the combination, using lentiviral transduction and in comparison with a scramble sequence. These were the ones most highly differentially expressed in aCM, and a quite significant lower level in iCM. Both miRNAs have been shown to foster myogenic differentiations of both ESCs and cardiovascular progenitors. Authors confirmed that both miRNAs enhanced myogenic commitment toward terminal differentiation of iCMs. At the end of the differentiation process direct targets of these two miRNAs involved in the cell cycle regulation, such as Cyclin D2 and Cdk6, for miR-1, and Ccna2 for miR-499 were down-regulated in engineered iCMs. The data showed that both, miR-1 and miR-499, regulate negatively the cell cycle gene network, which is typical of the myogenic terminal differentiation of cardiac muscle cells. In agreement with the central hypothesis, iCMs overexpressing with miR-1 and/or miR-499 show a significantly higher expression of Nkx2.5, Gata-4, Mef2c, Tbx5, Hand2, Myo6, Myo7, Tnnt3, Actc1 and Atp2a when compared to scrambled miRNA transduction. Interestingly, co-transduction with miR-1 and miR-499 further promoted the expression of Tbx5, Hand2, Myo6 and Tnnt3, while the expression of Tnnt2, Myl2, Myo7, and Atp2a was not further increased when compared with the transduction with only single miRNA. However some genes (Myl2, Myo7 and Tnnt2) were not affected by any of the single miRNAs or their combination. Overall, authors concluded that the large body of data supports the conclusion that the miRNA/mRNA network built by the bioinformatic analysis identified key miRNAs involved in myocyte commitment and terminal differentiation as indeed miR-1 and miR-499 were able to enhance myogenic commitment of CSC-derived iCMs toward the terminal differentiation and cell cycle withdrawal typical

of adult CMs. Therefore the results demonstrate that, under proper in vitro culture conditions, CSCs robustly differentiate into functional beating cardiomyocytes undergoing changes in their transcriptome/miRnome that reasonably resembles that of adult CMs, further supporting the true myogenic stem/precursor nature of CSCs. The work is interesting and of high intrinsic value for specialists in the field. The manuscript is clear and well-written, although with a high load of bioinformatics analysis. With the aim to improve the manuscript I would like to make the following comments.

Main comments

1. Authors indicate that they used cloned CSCs, but along the manuscript it is not indicated if the body of the results (except in some bioinformatic information) have been obtained from an individual clone or they are using a pool of different clones. Please indicate; this would be interesting. In any case, I think that some of the validation analyses could be done and presented for several independent clones. This would help to have a better idea of the homogeneity/heterogeneity of the differentiation response with the protocol described.

2. It will be also relevant that the validation experimentation with engineered CSCs to overexpress miR-1 or miR-499, or the combination, could be endorsed by in vivo data. Would be feasible to evaluate also in vivo the engineered iCMs compared with aCM properly labelled?

Minor points

1. Page 3; line 97. The adult cardiac stem/progenitor cells (CSCs) are----- (CSC/CPC)
2. Page 11; line 363. the transfection of miR-1 ---- transduction
3. Page 14. Line 437. Cardiac stem cells (CSCs)---- Cardiac Stem Cells or cardiac stem cells.
4. Page 14. Line 438. using gentleMACS. ---- gentle MACS
5. Page 29. Line 936. following transduction of iCMs ----- Following transduction in CSCs or following overexpression in iCMs
6. Figure 1. Include also the cTnI label in the panel 1D.

Reviewer #2 (Remarks to the Author):

This topic continues to be provocative and of broad interest. Therefore I recommend acceptance after my concerns have been resolved.

Reviewer #3 (Remarks to the Author):

Scalise et al. have taken clonally derived CSCs and differentiated them into contracting cardiomyocytes (iCMs) in vitro. The biochemical, structural and gene expression analyses of the iCMs resembles that of fetal to neonatal cardiomyocytes. RNAseq analysis revealed activation of the entire gene network characteristic of the adult cardiomyocyte phenotype. iCMs up-regulate known cardiomyo-miRs, but are significantly lower expression than adult CMs. Bioinformatics analysis predicted that CSC-iCM differentiation activates cardiomyocyte-specific and cell-cycle arrest gene

networks similar to those orchestrating cardiac development and maturation. miR-1 and miR-499 were able to enhance myogenic commitment of iCMs towards a terminal differentiation and cell cycle withdrawal typical of adult CMs.

The paper is very well written, clear and the data and its interpretation support the conclusions. I have the following minor comments:

Page 8, line 244, it states that 'Cluster 2 is a group of miRNAs (see Table S3) that are 245 significantly higher expressed in CSCs compared to the iCMs and aCMs (Figure 4B) and that 246 putatively target mRNAs involved with cell proliferation. Finally, the group organized as cluster 3 247 (Table S4) shows miRNAs up-regulated in iCMs compared to CSCs and aCMs (Figure 4B) that 248 putatively target mRNAs involved with positive stem cell population maintenance'. This to me seems odd because wouldn't the CSCs be expected to show upregulation of miRNAs that putatively target mRNAs involved in stem cell maintenance, rather than the iCMs, which should putatively target mRNAs involved with cell proliferation.

Page 11, line 338. Figure 7A, I do not see from the heatmap that mir-1-3p or mir-1-5p are also highly expressed in iCMs, like in aCMs. They show the lowest expression. mir-126-5p, mir335-3p and mir-335-5p are more highly expressed in iCMs.

Answers to Reviewer #1's comments:

The manuscript by Scalise et al. describes a well-defined *in vitro* procedure for the contractile cardiomyocyte differentiation (iCMs) from clonally-derived mouse CSCs. The protocol is similar to that described to generate cardiomyocytes from cardiovascular progenitors (CPC) from ESC-derived embryoid bodies (EB). iCM were nicely characterized and, in particular, it is described that dissociated iCMs at 14 days in differentiation medium showed a well-developed sarcomeric striations that more closely resembled neonatal cardiomyocytes (neoCMs isolated from 2 days old mice) which are still immature compared with that of adult terminally differentiated CMs (aCMs, isolated from 10 weeks old mice). RT-qPCR analysis shows that at the end of the differentiation protocol the main cardiac transcription factors, Gata-4, Nkx2.5, Mef2C and Hand2 were robustly up-regulated with concurrent expression of the contractile protein genes, such as cTnnt2, Myh7 and Actc1.

The RNASeq comparison of "CSCs vs. iCMs" mRNome and microRNome showed a balanced up-regulation of sarcomere and mitochondrial related mRNAs coupled to the a down-regulation of cell cycle and DNA replication mRNAs. Results indicated that the differentiation process is somehow incomplete not reaching the levels found in purified terminally differentiated adult cardiomyocytes (aCMs), showing intermediate levels between those of fetal and neonatal cardiomyocytes; this was confirmed by RT-qPCR analyses. In agreement, cardiomyo-miRs were clearly up-regulated in iCMs whereas those miRs positively regulating stem cell expansion and self-renewal were down-regulated. Again, the levels of regulation of key miRNAs are also lower in iCM compared with aCM. Therefore, authors conclude that the specific miRNA/mRNAs networks operative in iCMs closely resembled miRNA/mRNA networks of aCMs and proposed that the difference in intensity could be associated to the heterogeneity of the iCMs population. The integrated bioinformatics analysis yield compatible miRNA/mRNA interactions supporting the notion that CSC differentiation into iCMs activates cardiomyocyte-specific and cell-cycle arrest gene networks similar to those orchestrating cardiac development and maturation.

With no doubt, although ideally the described *in vitro* procedure for differentiation of contractile CM has a significant margin to be improved, it is the best procedure described using adult cardiac progenitors or CSCs.

As a validation for the differentiation protocol authors engineered CSC for overexpression of miR-1, miR-499, or the combination, using lentiviral transduction and in comparison with a scramble sequence. These were the ones most highly differentially expressed in aCM, and a quite significant lower level in iCM. Both miRNAs have been shown to foster myogenic differentiations of both ESCs and cardiovascular progenitors. Authors confirmed that both miRNAs enhanced myogenic commitment toward terminal differentiation of iCMs. At the end of the differentiation process direct targets of these two miRNAs involved in the cell cycle regulation, such as Cyclin D2 and Cdk6, for miR-1, and Ccna2 for miR-499 were down-regulated in engineered iCMs. The data showed that both, miR-1 and miR-499, regulate negatively the cell cycle gene network, which is typical of the myogenic terminal differentiation of cardiac muscle cells. In agreement with the central hypothesis, iCMs overexpressing with miR-1 and/or miR-499 show a significantly higher expression of Nkx2.5, Gata-4, Mef2c, Tbx5, Hand2, Myo6, Myo7, Tnnt3, Actc1 and Atp2a when compared to scrambled miRNA transduction. Interestingly, co-transduction with miR-1 and miR-499 further promoted the expression of Tbx5, Hand2, Myo6 and Tnnt3, while the expression of Tnnt2, Myl2, Myo7, and Atp2a was not further increased when compared with the transduction

with only single miRNA. However some genes (Myl2, Myo7 and Tnnt2) were not affected by any of the single miRNAs or their combination.

Overall, authors concluded that the large body of data supports the conclusion that the miRNA/mRNA network built by the bioinformatic analysis identified key miRNAs involved in myocyte commitment and terminal differentiation as indeed miR-1 and miR-499 were able to enhance myogenic commitment of CSC-derived iCMs toward the terminal differentiation and cell cycle withdrawal typical of adult CMs. Therefore, the results demonstrate that, under proper *in vitro* culture conditions, CSCs robustly differentiate into functional beating cardiomyocytes undergoing changes in their transcriptome/miRnome that reasonable resembles that of adult CMs, further supporting the true myogenic stem/precursor nature of CSCs. The work is interesting and of high intrinsic value for specialists in the field. The manuscript is clear and well-written, although with a high load of bioinformatics analysis.

With the aim to improve the manuscript I would like to make the following comments.

We thank the reviewer for such a detailed and thorough analysis of our work and for her/his complimentary comments to our study as well as for her/his advice/suggestions to improve our manuscript. We believe that we have satisfactorily answered each of her/his comments as described below.

Main comments

1. Authors indicate that they used cloned CSCs, but along the manuscript it is not indicated if the body of the results (except in some bioinformatic information) have been obtained from an individual clone or they are using a pool of different clones. Please indicate; this would be interesting. In any case, I think that some of the validation analyses could be done and presented for several independent clones. This would help to have a better idea of the homogeneity/heterogeneity of the differentiation response with the protocol described.

We thank the reviewer for addressing this important point. To obtain wild type CSC clones, 6 wild-type adult C57BL/6J mouse hearts were isolated and through negative/positive cell sorting, CD45^{neg}CD31^{neg}c-kit^{pos} cardiac cells were harvested and expanded in culture through 4 passages. At this point, single cells were deposited in 96 wells and 14 days after, 3 clones with the fastest expansion rate were picked. These 3 clones were then expanded for 4 passages. Because we have already shown that using this isolation method, the harvested clones behave practically in an indistinguishable manner both biologically as well as transcriptionally (Vicinanza et al. Cell Death and Differentiation 24, 2101–2116; 2017 doi:10.1038/cdd.2017.130), only one randomly picked clone was used to perform the analysis in the present study whereby biological replicates of the same CSC clone were used for the analysis of the process of cardiomyocyte differentiation. Following the reviewer's comment, because data were already available, we have added the characterization of myocyte commitment from the other two additional clones. When comparing the level of myogenic commitment in the differentiation assay *in vitro* of these two different clones with the original one, it is evident that they all reached similar myocyte differentiation levels when evaluated by cardiac transcription factor and contractile gene mRNA expression and myofilament contractile protein expression (see new Supplementary Figure 1). In short, each set of results presented were obtained from single clone-derived cells and confirmed by analyses of cells derived from two other clones.

2. It will be also relevant that the validation experimentation with engineered CSCs to overexpress miR-1 or miR-499, or the combination, could be endorsed by in vivo data. Would be feasible to evaluate also in vivo the engineered iCMs compared with aCM properly labelled?

We thank the reviewer for such a comment. We hope that s/he understands that to fully address such a comment it would require an extensive number of new experiments whereby these new data could stand as a separate study on its own. Indeed, the Editor fairly noted that *“the request to show in vivo validation is optional, as it can be viewed as outside of the scope of this manuscript”*. We therefore hope that Reviewer #1 accepts this view even though it is clear to these authors that *in vivo* validation of miR modulation for cardiomyocyte maturation from CSC differentiation is a key point to address in the next future as the work is already in progress. We have added this limitation in the Discussion section of our revised manuscript (see lines 448-451).

Minor points

1. Page 3; line 97. The adult cardiac stem/progenitor cells (CSCs) are----- (CSC/CPC)

Thanks, done.

2. Pag 11; line 363. the transfection of miR-1 ---- transduction

Thanks, done.

3. Pag 14. Line 437. Cardiac stem cells (CSCs)---- Cardiac Stem Cells or cardiac stem cells.

Thanks, done.

4. Pag 14. Line 438. using gentleMACS. ---- gentle MACS

Thanks, the gentleMACS is the brand name used by the manufacturer.

5. Pag 29. Line 936. following transduction of iCMs ----- Following transduction in CSCs or following overexpression in iCMs

Thanks, done.

6. Figure 1. Include also the cTnI label in the panel 1D.

Thanks, done.

Answers to Reviewer #2's comments:

An interesting and worthwhile study of CSC commitment. The authors are one of the only groups in the world capable of this in-depth assessment and understanding of CSC biological properties. Being an *in vitro* study, this manuscript has focused upon the potential of CSC differentiation and avoided the highly controversial topic of CSC contribution to cardiomyogenesis *in vivo*. Nevertheless, the Discussion is currently rather brief and would benefit from a brief exploration of relevance to *in vivo* cardiomyocyte generation / replacement by CSC. The authors are leading proponents for CSC cardiomyogenesis in myocardial biology, so it would be appropriate and reasonable to extrapolate from their directed differentiation model system *in vitro* into how efficiently and frequently such transitions from CSC to cardiomyocytes occur *in vivo*. I would not be surprised to find out that the authors hold the minority opinion on this topic. Therefore, this reviewer urges the authors to present a balanced perspective that addresses their findings as well as that of skeptics who remain unconvinced of (or have decided against) a role for CSC cardiomyogenesis *in vivo*. This should not dominate the Discussion since the authors have written other papers on this topic, but their prior publications addressing this concern ought to be contextualized and referenced here.

We thank the reviewer for her/his complimentary comments for our study as well as for her/his advice/suggestions to improve our manuscript and wade again into the controversial aspects of this field, something we carefully avoided in the original manuscript. However, we welcome the opportunity to present our point of view yet again, which in this case is indirectly supported by the *in vitro* data. Accordingly, we have revised the Discussion to present a balanced perspective on the role of CSC cardiomyogenesis *in vivo* (see lines 421-451). Furthermore, we believe that we have satisfactorily addressed each of her/his comments as described below.

Additionally, I offer the following observations on the figures and their interpretation that would benefit from further explanation and refinement:

- Scale bars need to be legible.

Thanks, done.

- Fluorescent targets need to be labeled in all micrographs (F1)

Thanks, done

- The use of heatmaps and bar graphs to represent the same data in Figure 2, 3F and 7 is redundant and could be simplified.

We understand the comment and agree that they represent the same data. However, we hope that the reviewer would agree with us that an approach of '*repetita iuvant*' is the best to facilitate readers to fully appreciate the difference in gene expression levels. Therefore, respectfully we decided to leave the images as they are.

- Gene expression and enrichment graphics need to increase in resolution and clearly display the information presented in the written narrative of the results. Figures 3(a-e), 4(a,e,g and i), 5 and 6 results are not legible.

We thank the reviewer and addressed this comment by providing vectorial images that by default have the highest possible resolution. This will enable reader to clearly read by zooming in the image.

- Fig 4 - k-means clustering should be outlined on panel a heatmap and panel b should indicate what exactly is the y-axis measuring.

Thanks, done.

- Fig 4d suggest that iCMs correlate closer to CSC rather than adult CMs which runs opposite to the title. Would it be better to simply drop the word "adult" and then assert that CSC acquire the blueprint of cardiomyocytes (maturation level notwithstanding). I don't think anyone would be surprised to see that directed differentiation of CSC in vitro yields a relatively immature cardiomyocyte phenotype.

Thanks, the title of the ms reflects now what it is suggested by Reviewer #2.

- In Fig 4(f,h) BinGO output similar biological processes cell differentiation and heart development on both pairwise comparisons. It is difficult to derive biological meaning when it is unknown which miRNAs are overlapping between CSC, iCMs and aCMs. Also the BinGO input seems to be global miRNA differential expression but stratified in up and down regulation, which may yield more biological meaning.

Thanks, done. Please see new Figure 4f, h.

- Dot plots or violin plots overlaid on the boxplots on Fig 4(b,j) would represent miRNA changes better

Thanks, done. Please see new Figure 4b, j.

- Fig 5 and 6 are not readily legible and while network analysis is important the biological information of these figures is lost on the chosen representation. Targets and miRNAs cannot be easily identified.”

We thank the reviewer and addressed this comment providing vectorial images that by default have the highest possible resolution. This will enable reader to clearly read by zooming in the image. Furthermore, to have direct appreciation of the networks we created four new tables, respectively two for figure 5 and two for figure 6, where the reader will be able to assess the interaction of the miRNAs with their specific targets (see new Supplementary Table 7,8,9,10).

Specific Answers to Reviewer #3's Comments:

Scalise et al. have taken clonally derived CSCs and differentiated them into contracting cardiomyocytes (iCMs) *in vitro*. The biochemical, structural and gene expression analyses of the iCMs resembles that of fetal to neonatal cardiomyocytes. RNAseq analysis revealed activation of the entire gene network characteristic of the adult cardiomyocyte phenotype. iCMs up-regulate known cardiomyo-miRs, but are significantly lower expression than adult CMs. Bioinformatics analysis predicted that CSC-iCM differentiation activates cardiomyocyte-specific and cell-cycle arrest gene networks similar to those orchestrating cardiac development and maturation. miR-1 and miR-499 were able to enhance myogenic commitment of iCMs towards a terminal differentiation and cell cycle withdrawal typical of adult CMs.

The paper is very well written, clear and the data and its interpretation support the conclusions. I have the following minor comments.

We thank the reviewer for her/his complimentary comments as well as for her/his advice/suggestions to improve our manuscript. We believe that we have satisfactorily addressed each of her/his comments as described below.

Page 8, line 244, it states that 'Cluster 2 is a group of miRNAs (see Table S3) that are significantly higher expressed in CSCs compared to the iCMs and aCMs (Figure 4B) and that putatively target mRNAs involved with cell proliferation. Finally, the group organized as cluster 3 (Table S4) shows miRNAs up-regulated in iCMs compared to CSCs and aCMs (Figure 4B) that putatively target mRNAs involved with positive stem cell population maintenance'. This to me seems odd because wouldn't the CSCs be expected to show upregulation of miRNAs that putatively target mRNAs involved in stem cell maintenance, rather than the iCMs, which should putatively target mRNAs involved with cell proliferation.

We understand the concern and expectation of the Reviewer #3. His/her concern might be mainly due to the nomenclature of the clustering program, whereby miRNA clustering (obtained from the comparison of the 3 different cell population for each of the analysis) clusters together microRNAs whose putative mRNA targets are involved in a particular biological function without distinguishing whether their role is either positive or negative. Thus, when referring to cell cycle, this indicates the biological function, and cluster together both positive and negative regulator genes of the cell cycle. Because of this *a priori*, it is not surprising that clonal CSCs, being highly proliferating, shows an upregulation of microRNA involved with cell cycle modulation as main biological function. Concurrently, the same explanation is valid for cluster 3 grouping microRNAs involved with stem cell maintenance.

Page 11, line 338. Figure 7A, I do not see from the heatmap that mir-1-3p or mir-1-5p are also highly expressed in iCMs, like in aCMs. They show the lowest expression. mir-126-5p, mir335-3p and mir-335-5p are more highly expressed in iCMs.

We again understand the reviewer concern. However, the heatmap clearly shows that miR-1 and miR-499 were the ones to have the highest expression in aCMs and were also highly expressed in the iCMs (when compared to CSCs) albeit at a significantly lower levels than in the aCMs. Thus, miR-1 and miR-499 were chosen for overexpression experiments to test their role in maturation of iCMs.

REVIEWERS' COMMENTS:

Reviewer #1 (Remarks to the Author):

I agree with the revised version submitted and the answers provided. In addition, I understand that further in vivo evaluation of genetically engineered CSCs would better fit within a new whole manuscript.

Reviewer #2 (Remarks to the Author):

All my prior concerns have been appropriately addressed by the authors in their revision. No further concerns.

Reviewer #3 (Remarks to the Author):

The authors have addressed all my comments.